# Social reward outcompetes drug seeking dopaminergic ensembles to prevent relapse

Wei Zheng [1,11], Xiaoxing Liu [2,11], Tangsheng Lu[3,11], Xinyou Lv[4], Xuefang Guan [5], Yifan Yu [2], Xue Li[3], Zhe Wang[6], Kai Yuan[2], Jeffrey W. Grimm[7], Trevor W. Robbins [8,9], Jie Shi [3] ✉, Lin Lu [1,2,3,6] ✉ & Yan-Xue Xue [3,10] ✉

Drugs of abuse promote substance use disorder (SUD) by hijacking meso-limbic circuits that normally process natural rewards. Among these, social rewards exhibit therapeutic potential, but the underlying neural substrates remain unclear. Using a multimodal approach integrating in vivo single-neuron calcium imaging, optogenetic manipulation, and electrophysiology in male rats, we identified two distinct dopaminergic ensembles in the ventral tegmental area (VTA) that respectively encode social reward and drug seeking. Notably, these antagonistic ensembles exert reciprocal influence through competitive interactions that shape behavioral outcomes. Furthermore, circuit mapping revealed divergent connectivity patterns, with social reward-responsive dopaminergic ensembles receiving preferential input from the dorsal raphe nucleus (DRN). Activation of the DRN-VTA pathway recapitulates the protective effects of social reward against drug seeking. In this study, we uncovered a dynamic competition between functionally specialized dopaminergic ensembles through which social reward attenuates drug seeking, offering insights that may inform development of novel strategies for SUD treatment.

Substance use disorder (SUD) is a chronic, relapsing brain disorder that poses a significant global health challenge[1]. Extensive research has demonstrated that drugs of abuse hijack brain reward systems that typically regulate the pursuit of natural rewards, resulting in maladaptive drug-seeking behaviors[2,3]. However, recent evidence suggests that natural rewards, particularly social rewards[4], can counteract the impact of drugs and provide protection against SUD. Positive social interactions are increasingly recognized as effective alternatives to addictive substances, offering potential interventions in critical phases of SUD, including initiation[5], extinction[6], and incubation[7]. While social-based approaches are cost-effective and associated with minimal adverse effects, their neurobiological mechanisms remain elusive.

One region of interest is the ventral tegmental area (VTA), where dopaminergic (DAergic) neurons coordinate the reinforcing properties of both social reward[8–10] and drugs of abuse[11,12]. Growing evidence indicates substantial functional heterogeneity within VTA DAergic

[1]Peking-Tsinghua Center for Life Sciences, PKU-IDG/McGovern Institute for Brain Research, Peking University, Beijing, China. [2]Peking University Sixth Hospital, Peking University Institute of Mental Health, NHC Key Laboratory of Mental Health (Peking University), National Clinical Research Center for Mental Disorders (Peking University Sixth Hospital), Beijing, China. [3]National Institute on Drug Dependence, Beijing Key Laboratory of Drug Dependence, Peking University, Beijing, China. [4]Department of Psychology, School of Humanities and Social Sciences, University of Science and Technology of China, Hefei, China. [5]Henan Academy of Innovation in Medical Science, Henan University, Kaifeng, China. [6]Institute of Brain Science and Brain-inspired Research, Shandong First Medical University and Shandong Academy of Medical Sciences, Jinan, China. [7]Department of Psychology and Program in Behavioral Neuroscience, Western Washington University, Bellingham, WA, USA. [8]Behavioural and Clinical Neuroscience Institute, Department of Psychology, University of Cambridge, Cambridge, UK. [9]Institute of Science and Technology for Brain-Inspired Intelligence, MOE Frontiers Center for Brain Science, Fudan University, Shanghai, China. [10]Chinese Institute for Brain Research, Beijing, Beijing, China. [11]These authors contributed equally: Wei Zheng, Xiaoxing Liu, Tangsheng Lu. ✉e-mail: shijie@bjmu.edu.cn; linlu@bjmu.edu.cn; yanxuexue@bjmu.edu.cn

subpopulations, characterized by distinct cellular morphologies, neurochemical profiles, electrophysiological properties, and differential responses to rewarding versus aversive stimuli[13,14]. This diversity raises the possibility of specialized DAergic populations selectively encoding social reward versus drug-related behaviors. Paradoxically, there is evidence of significant overlap among VTA DAergic neurons in response to food and social interaction[10], as well as in nucleus accumbens (NAc) medium spiny neurons (MSNs) responding to both food/drink and addictive substances[2]. These contradictory observations create a fundamental tension in addiction neuroscience: are there segregated DAergic ensembles in the VTA dedicated to social versus drug-related behaviors, or do overlapping neuronal populations mediate both?

In this work, we perform in vivo calcium imaging to identify distinct DAergic ensembles in the VTA that are responsive to social reward (Social-reward-Ens[DA]) and drug seeking (Drug-seeking-Ens[DA]). Multimodal interrogation combining optogenetic manipulation with electrophysiology reveals dynamic, reciprocal inhibition between these two functionally segregated populations. Circuit tracing and chemogenetic perturbation further delineate an organized circuit from the dorsal raphe nucleus (DRN) to VTA that selectively modulates the protective effect of social reward against drug seeking. Together, these findings establish both cellular and circuit-level mechanisms by which socially engaged DAergic ensembles constrain drug-seeking-associated neural plasticity, revealing potential therapeutic targets for preventing relapse.

## Results

### VTA DAergic neurons as a central nexus of social reward and drug seeking

Previous studies have shown that interacting with juvenile conspecifics can serve as an experimental manipulation of social reward in mice[15], with mice developing a preference for a compartment associated with the presence of two juvenile conspecifics[16]. In the present study, we investigate whether this similar social paradigm can serve as a social reward manipulation in rats. Behavioral assays revealed that rats exhibited a robust conditioned place preference for the context paired with juvenile conspecifics (Supplementary Fig. 1a–d) and successfully acquired the operant response required to access juvenile conspecifics (Supplementary Fig. 1e, f). Furthermore, activating the neurons activated by social interaction with juvenile conspecifics induces intracranial self-stimulation behavior (Supplementary Fig. 1g–j). Collectively, these findings confirm that interaction with juvenile conspecifics can serve as a valid experimental manipulation of social reward in rats. Next, we assessed whether social reward could suppress cocaine (a psychostimulant) or heroin (an opioid) seeking during the relapse test. As in previous work[17], rats underwent a cocaine or heroin relapse test following 10 days of self-administration training (cocaine or heroin was injected intravenously and paired with a light-tone cue after the rat nose-poked in the active hole), 14 or 7 days of self-administration extinction training (only a light-tone cue was delivered after the rat nose-poked in the active hole), and 4 weeks of home cage recovery (Fig. 1a–c and Supplementary Fig. 2a–c). The relapse test was manifested by cue-induced active nose-pokes for drug seeking. Before the drug relapse test, rats were divided into two groups: a social reward group and a no-reward group. Rats in the social reward group were placed in a box and allowed to freely interact with two juvenile conspecifics for 30 min before the relapse test, while rats in the no reward group were placed alone in the same box for 30 min (Fig. 1a and Supplementary Fig. 2a). Social reward significantly reduced cue-induced active nose-pokes both in the cocaine (Fig. 1d) and heroin relapse tests (Supplementary Fig. 2d), with no effect on inactive pokes, suggesting a selective suppression of drug seeking. To ensure that the observed effect was due to social reward, rather than novelty exploration, two control groups were included: rats were exposed to

either two toy rats or anesthetized conspecifics for 30 min prior to the relapse test (Supplementary Fig. 2e–g, i–k). Neither of these manipulations affected nose-poke behaviors (Supplementary Fig. 2h, l), confirming that the suppression of drug seeking was driven by social reward, not novelty per se.

The VTA DAergic neurons mediate the rewarding properties of both social interaction[8–10] and drugs of abuse[11,12]. Thus, we attempted to dissect the role of VTA DAergic neurons in mediating the effect of social reward against drug seeking. To achieve this, we used adeno-associated virus (AAV)-tyrosine hydroxylase (TH)-GCamp6s to specifically target VTA DAergic neurons (Fig. 1e). GCamp6s expression was restricted to the VTA, with no spread to adjacent regions (Fig. 1f). Calcium activity was monitored via fiber photometry during both social interaction and the cocaine relapse test (Fig. 1g). Body contact during social interaction reliably evoked robust calcium transients in VTA DAergic neurons (Fig. 1h–j). Similarly, drug seeking during the relapse test was associated with a marked increase in calcium signal (Fig. 1k–m). Notably, this drug-seeking-associated activity was significantly attenuated in rats that had prior exposure to social interaction (Fig. 1n–r), suggesting that VTA DAergic neurons contribute to the suppressive effects of social reward on drug seeking.

To further examine changes in DAergic neurotransmission, we assessed DA release in the NAc, the primary target of monosynaptic output from VTA DAergic neurons[18]. A DA sensor was injected into the NAc to enable fiber photometry recordings (Supplementary Fig. 3a–c). Body contact during social interaction induced a notable increase in DA release within the NAc (Supplementary Fig. 3d–f). Similarly, drug seeking during the relapse test reliably increased DA release in the NAc (Supplementary Fig. 3g–i). However, this drug-seeking-induced increase in DA release was significantly attenuated after social interaction (Supplementary Fig. 3j–n), suggesting that social reward decreased DA release following drug seeking.

### Distinct DAergic ensembles respond to social reward and drug seeking

While classic work on DAergic neurons in the VTA has highlighted their apparent homogeneous responses to various stimuli[19], recent research has uncovered notable heterogeneity at the single-cell level[20]. For instance, 15 distinct DAergic subpopulations have been identified based on molecular expression profiles, with these subpopulations displaying diverse responses to rewards and aversive events[13].

To determine whether distinct or overlapping DAergic subpopulations respond to social reward and drug seeking, we employed an activity-dependent dual-labeling approach. Specifically, we used an AAV expressing ERCreER under the control of an activity-dependent promoter (Label 1, AAV-cfos-ERCreER-PEST, hereafter referred to as TRAPed neurons) and immunofluorescent detection of cfos expression (Label 2) to label neurons activated by social interaction and/or cocaine seeking during the relapse test (Fig. 2a, b). To validate the TRAPing paradigm in rats, we first characterized the temporal dynamics of neuronal labeling. Rats were subjected to 30 min of social interaction at varying time intervals relative to the administration of 4-hydroxytamoxifen (4-OHT). Analysis of mCherry expression patterns demonstrated that the majority of TRAPed neurons were labeled within a 6-h time window centered on 4-OHT injection. Notably, mCherry expression levels peaked when 4-OHT was administered 30 min prior to the initiation of social interaction (Supplementary Fig. 4a–d). To control for potential confounding effects of viral infection, rats were assigned to four groups: Social-reward (Label 1) + Drug-seeking (Label 2), Drug-seeking (Label 1) + Social-reward (Label 2), Drug-seeking (Label 1) + Drug-seeking (Label 2), and Social-reward (Label 1) + Social-reward (Label 2) (Fig. 2c and Supplementary Fig. 5a, b). DAergic neurons were identified by immunostaining for tyrosine hydroxylase (TH) (Fig. 2d, and Supplementary Fig. 5c). Quantification revealed that co-labeled neurons—responsive to both drug seeking

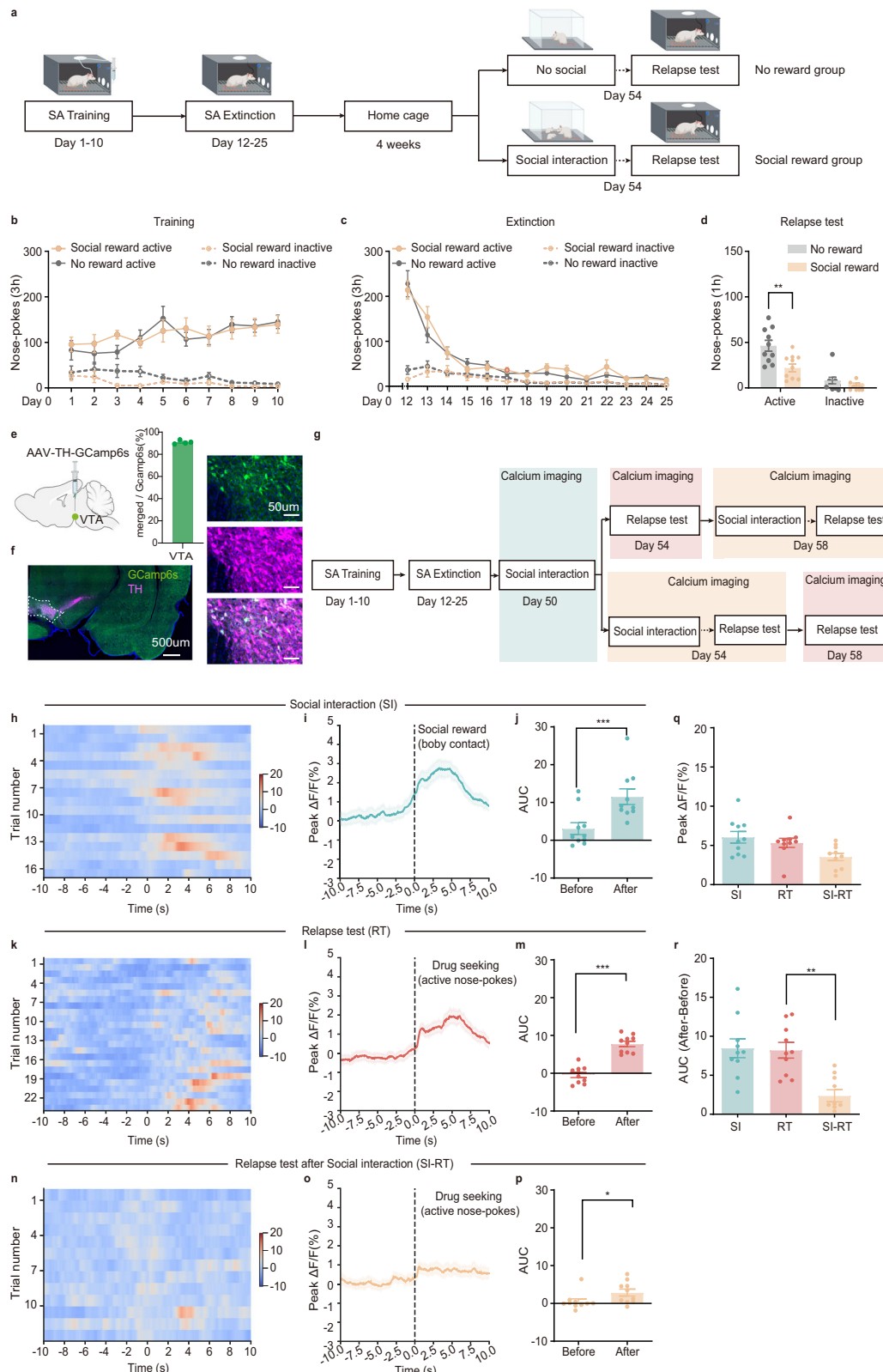

and social reward—comprised only ~13.35% of all Drug-seeking-responsive neurons and ~15.58% of all Social-reward-responsive neurons (Supplementary Fig. 5d–i), and co-labeled DAergic neurons—responsive to both drug seeking and social reward—comprised only ~8.93% of all Drug-seeking-responsive DAergic neurons and ~10.55% of all Social-reward-responsive DAergic neurons (Supplementary Fig. 5d–i). Notably, the proportion of co-labeled DAergic neurons

across social reward and drug seeking conditions was significantly lower than that observed in control groups subjected to the same stimulus twice (Fig. 2e, f), suggesting largely distinct neuronal populations for each condition.

Neuronal ensembles are defined as populations of neurons that encode specific information through coordinated activity patterns induced and shaped by learning[21]. To determine whether the TRAPed

**Fig. 1 | The VTA DAergic neurons act as a central nexus of social reward and drug seeking. a** Schematic of the experimental design. **b** Number of nose-pokes during cocaine self-administration training ($n = 10$ male rats, three-way ANOVA, 10 consecutive days measurements from the same animal: $F_{(9,10)} = 2.799$, $p = 0.062$; see Supplementary Table 1A). **c** Number of nose-pokes during extinction ($n = 10$ male rats, three-way ANOVA, 14 consecutive days measurements from the same animal: $F_{(13,6)} = 1.233$, $p = 0.425$; see Supplementary Table 1B). **d** Number of nose-pokes during the relapse test, showing that social reward significantly inhibits cue-induced active nose-pokes ($n = 10$ male rats, two-way ANOVA: $F_{(1,18)} = 5.857$, $p = 0.026$; see Supplementary Table 1C). **e, f** Representative photo of GCamp6s expression in the VTA ($n = 4$ male rats, repeated 4 times independently with similar results). Green: GCamp6s. Pink: Tyrosine hydroxylase, TH. **g** Schematic of fiber photometry recording. **h, k, n** Trial-by-trial heatmap representations of calcium signals. Color scales on the right indicate $\Delta F/F$. **i, l, o** Event plot of average calcium signals aligned to distinct behavioral manipulation. **j, m, p** Comparison of the area under curve (AUC) between baseline (10 s before stimuli) and stimuli (within 10 s from onset of stimuli) ($n = 10$ male rats, paired two-tailed t test: SI, $t_{(9)} = 7.004$, $p < 0.001$; RT, $t_{(9)} = 8.131$, $p < 0.001$; SI-RT, $t_{(9)} = 3.043$, $p = 0.014$). **q** Peak $\Delta F/F$ quantification ($n = 10$ male rats, one-way ANOVA with Bonferroni multiple comparisons test, $F_{(2,18)} = 4.765$, $p = 0.022$; RT vs SI-RT, $p = 0.109$). **r** AUC quantification ($n = 10$ male rats, one-way ANOVA with Bonferroni multiple comparisons test, $F_{(2,18)} = 11.08$, $p = 0.007$; RT vs. SI-RT, $p = 0.002$). All the data are presented as mean ± SEM. 95% confidence interval was used for all statistical analyses in this figure. **a, e**: Created in BioRender. Xiaoxing, L. (2026) https://BioRender.com/hqs0eg6. $^*p < 0.05$, $^{**}p < 0.01$, $^{***}p < 0.001$.

neurons in the VTA constitute a neuronal ensemble, we examined whether neurons labeled during the first episode of drug seeking or social interaction could be reactivated during a second, temporally separated episode of the same stimulus. Indeed, re-exposure to the same condition robustly reactivated TRAPed neurons, as evidenced by their overlap with cfos expression, confirming the presence of stable drug-seeking- and social-reward-responsive ensembles within the VTA (Supplementary Fig. 5j–o). Finally, spatial mapping of DAergic ensembles revealed that both social-reward- and drug seeking-responsive ensembles were preferentially localized to the middle VTA (AP −5.4 to −5.8 mm), with no significant spatial segregation between the two (Supplementary Fig. 6a–c). Moreover, the number of DAergic neurons activated during social reward versus drug relapse sessions did not differ significantly; both populations accounted for approximately 30% of total DAergic neurons in the VTA (Supplementary Fig. 6d, e). Collectively, these findings demonstrate the existence of functionally distinct DAergic ensembles that selectively encode social reward and drug seeking.

To further confirm that distinct ensembles encode social reward and drug seeking, we tracked VTA DAergic neurons' activity at single-cell resolution using single-photon calcium imaging in free-moving rats (Fig. 2g). AAV-TH-GCamp6s was injected into the VTA, and gradient-index (GRIN) lenses were implanted after a 3-week recovery (Fig. 2h). This approach enabled direct comparison of neuronal responses to both stimuli within the same imaging field. We identified discrete populations of VTA DAergic neurons responsive to either cocaine seeking, social interaction, or both (Fig. 2i–t). Across all animals, 948 neurons were extracted, of which 380 were successfully aligned across the two sessions (Fig. 2u). Among these, 22% (84 out of 380 neurons, $n = 6$ animals) responded selectively to cocaine seeking during the relapse test (Figs. 2i–k, v), 17% (65 out of 380 neurons, $n = 6$ animals) responded selectively to body contact during social interaction (Figs. 2l–n, v), and only 9% (34 out of 380 neurons, $n = 6$ animals) responded to both (Fig. 2o–q, v). Statistical analysis revealed no significant overlap between Drug-seeking- and Social-reward-responsive neurons (Fig. 2w). If social reward and drug seeking are encoded by distinct neuronal populations, their response magnitudes across the population would be expected to exhibit no correlation or a negative correlation. Consistent with this, we observed a significant inverse correlation between neuronal responses to social interaction and cocaine seeking across the VTA DAergic population (Fig. 2x, $r = −0.48$, $p < 0.01$). Together, these results indicate that most VTA DAergic neurons responsive to cocaine seeking are unresponsive to social interaction, supporting the existence of distinct DAergic ensembles encoding social reward and drug seeking.

To rule out the possibility that the segregation was merely due to stimulus presentation at distinct time points, we tracked VTA DAergic neuronal activity across two separate sessions of cocaine seeking and/or social interaction conducted on different days (Supplementary Figs. 7a, b and 8a, b). We found a significant overlap between the neuronal populations responsive to the same stimulus across sessions.

Specifically, 62% of VTA DAergic neurons responsive to the first cocaine seeking session were reactivated during the second drug seeking session (Supplementary Fig. 7c–r), and 56% of those responsive to the first social reward session were reactivated during the second social reward session (Supplementary Fig. 8c–r). In addition, given the substantial array of confounding variables inherent to the social interaction versus drug relapse sessions, one might hypothesize that the observed segregation of neuronal populations stems from divergent sensory and behavioral modalities. To address this alternative explanation, we trained rats to perform operant responses for both social self-administration and cocaine self-administration, then assessed whether distinct neuronal ensembles shaped by learning were engaged when rats executed goal-directed behaviors to obtain these two discrete reward-related stimuli (Supplementary Fig. 9a, b). Our results demonstrated that only 10% of recorded neurons (26 out of 261 neurons; $n = 2$ animals) exhibited dual responsiveness to both social and cocaine cues (Supplementary Fig. 9c–p). Subsequent statistical analyses confirmed a lack of significant overlap between social-seeking- and drug-seeking-responsive neuronal populations (Supplementary Fig. 9q). Collectively, these findings establish that VTA DAergic populations exhibit stable, stimulus-specific identities and that their segregation is not a consequence of temporal factors or behavioral confounds, consistent with the properties of learning-induced neuronal ensembles.

## Reciprocal inhibition of Social-reward-Ens$^{DA}$ and Drug-seeking-Ens$^{DA}$

Next, we investigated the relationship between Social-reward-Ens$^{DA}$ and Drug-seeking-Ens$^{DA}$. Previous studies have shown that midbrain DAergic neurons can exert inhibitory effects on downstream targets via non-canonical release of γ-aminobutyric acid (GABA)[22,23]. Based on this, we hypothesized that Social-reward-Ens$^{DA}$ may inhibit Drug-seeking-Ens$^{DA}$ through GABAergic transmission.

To test this hypothesis, we expressed the light-activated cation channel channelrhodopsin-2 (ChR2) in Social-reward-Ens$^{DA}$ using Flp recombinase-dependent AAVs (Label 1) and performed whole-cell voltage-clamp recordings from Drug-seeking-Ens$^{DA}$ (Label 2) in sagittal brain slices of VTA (Fig. 3a–d). Optogenetic stimulation of Social-reward-Ens$^{DA}$ with brief (1 ms) flashes of blue light reliably evoked large outward currents in ~72% of tested Drug-seeking-Ens$^{DA}$ ($n = 4$ animals, 18/25 neurons; Fig. 3e). These currents were abolished by the voltage-gated Na$^+$ channel blocker tetrodotoxin (TTX) and rescued by co-application of TTX and the K$^+$ channel blocker 4-aminopyridine (4-AP), consistent with monosynaptic connectivity ($n = 4$ animals, 18 neurons; Fig. 3e). Furthermore, pharmacological profiling revealed that these light-evoked currents were blocked by GABA$_A$ receptor antagonist picrotoxinin (PTX; $n = 3$ animals, 13 neurons, Fig. 3f), but were unaffected by the AMPA receptor antagonist NBQX ($n = 3$ animals, 11 neurons; Fig. 3g) or D1 receptor antagonist SCH-23390 ($n = 3$ animals, 10 neurons; Fig. 3h), confirming that the inhibitory postsynaptic currents (IPSCs) were mediated by GABA$_A$ receptors. This inhibitory influence

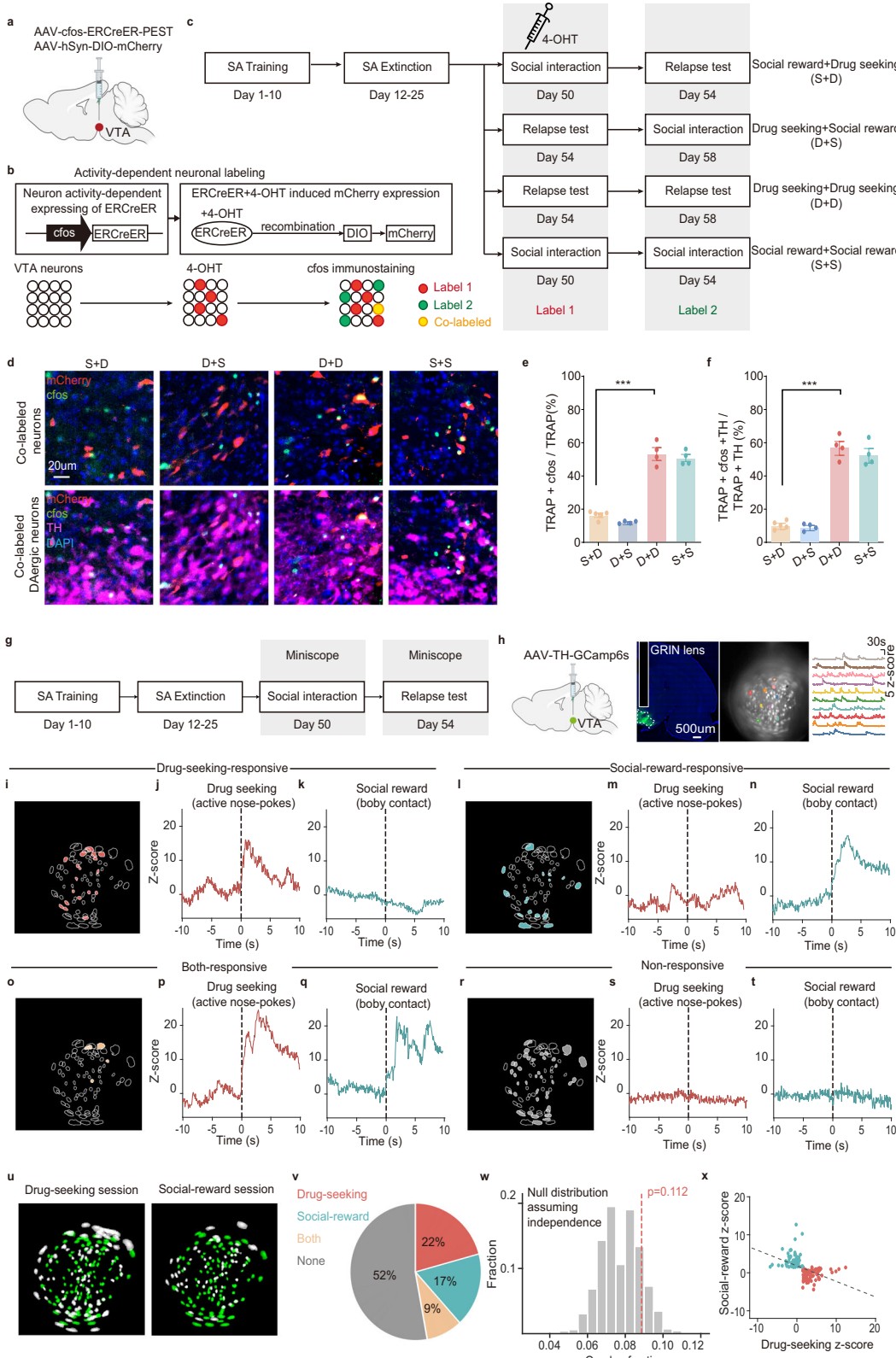

was highly specific. Non-specific optogenetic activation of random DAergic neurons failed to elicit IPSCs in ~87% of Drug-seeking-Ens$^{DA}$ in the VTA ($n = 3$ animals, 13/15 neurons; Supplementary Fig. 10a–e). To test whether the synaptic effects of Social-reward-Ens$^{DA}$ terminals could be sustained at high frequency, we performed cell-attached recordings from Drug-seeking-Ens$^{DA}$. High-frequency photostimulation (40 Hz, 1 min) of Social-reward-Ens$^{DA}$ terminals persistently

decreased the firing rate of ~64% of tested Drug-seeking-Ens$^{DA}$ ($n = 3$ animals, 9/14 neurons; Fig. 3i), suggesting a net inhibitory influence. These results establish a specific, functional inhibition from Social-reward-Ens$^{DA}$ to Drug-seeking-Ens$^{DA}$.

Given that previous studies demonstrated that Aldehyde Dehydrogenase 1a1 (ALDH1a1) but not the glutamate decarboxylase (GAD) mediates a GABA synthesis pathway in midbrain DAergic neurons[22,23],

**Fig. 2 | Distinct VTA DAergic ensembles respond to social reward and drug seeking. a**, **b** Activity-dependent neuronal labeling strategy. **c** Schematic of neuronal labeling (Label 1) and immunofluorescence staining (Label 2). **d** Representative tyrosine hydroxylase (TH) staining images (red: mCherry; green: cfos; pink: TH; blue: DAPI; scale bar: 20 µm); **e** Proportion of co-labeled neurons (S + D, $n = 5$ male rats; D + S, D + D and S + S, $n = 4$ male rats; one-way ANOVA with Bonferroni multiple comparisons test, $F_{(3,13)} = 117.0$, $p < 0.001$; S + D vs. D + D, $p < 0.001$. **f** Proportion of neurons co-labeled for Label 1, Label 2, and TH (S + D, $n = 5$ male rats; D + S, D + D, and S + S, $n = 4$ male rats; one-way ANOVA with Bonferroni multiple comparisons test, $F_{(3,13)} = 109.3$, $p < 0.001$; S + D vs D + D, $p < 0.001$. **g** Schematic of microscope calcium imaging. **h** Virus expression and GRIN lens placement (scale bar: 500 µm; repeated 4 times independently with similar results). **i**, **l**, **o**, **r** Example field of view with neurons colored by the stimulus type that they are responsive to. **j**, **k** Example of a neuron selectivity responsive to cocaine seeking. **m**, **n** Example of a neuron selectivity responsive to social reward. **p**, **q** Example of a neuron responsive to both. **s**, **t** Example of a neuron responsive to neither. **u** Matching neurons imaged across distinct sessions ($n = 380/948$ neurons). **v** Percentage of neurons responsive to social reward, cocaine seeking, both, or neither ($n = 380$ neurons). **w** Overlap relative to null distribution ($n = 380$ neurons, nonparametric permutation test, one-tailed, $p = 0.112$). **x** Correlation across neurons' activity in response to social reward and cocaine seeking ($n = 129$ neurons, Pearson's correlation, $r = -0.4843$, $p < 0.001$); neurons colored by selectivity to social reward and cocaine seeking. All the data are presented as mean ± SEM. 95% confidence interval was used for all statistical analyses in this figure. **a**, **h**: Created in BioRender. Xiaoxing, L. (2026) https://BioRender.com/hqs0eg6. $^*p < 0.05$, $^{**}p < 0.01$, $^{***}p < 0.001$.

we sought to determine whether ALDH1a1 drives functional GABA co-release by Social-reward-Ens[DA]. To address this, we examined GAD and ALDH1a1 expression in Social-reward-Ens[DA] via immunohistochemical co-labeling with TH. Our results revealed that only ~10% of Social-reward-Ens[DA] expressed GAD, whereas ~37% of these neurons expressed ALDH1a1 (Supplementary Fig. 11a–d). Moreover, treatment with the ALDH inhibitor 4-(diethylamino)-benzaldehyde (DEAB, 10 µM) but not the GAD inhibitor 3-mercaptopropionic acid (3-MPA, 500 µM) dramatically reduced light-induced IPSC amplitude (Supplementary Fig. 11e–i). Collectively, these findings establish that ALDH1a1 is required for functional GABA co-release by Social-reward-Ens[DA].

Based on this circuit-level inhibition, we hypothesized that social interaction engaging Social-reward-Ens[DA] would reduce the responsiveness of Drug-seeking-Ens[DA]. We tested this by longitudinally imaging the same DAergic populations across different social conditions (social reward or no social reward) (Fig. 3j). Indeed, following a 30-min free social interaction, the response of Drug-seeking-Ens[DA] was markedly attenuated, with both the magnitude and proportion of DAergic neurons responsive to drug seeking significantly diminished ($n = 5$ animals, 110 neurons; Fig. 3k–o). This effect was specific to the Drug-seeking-Ens[DA], as neurons responsive to social reward or those responsive to both categories were unaffected (Fig. 3p, q). Importantly, control animals without prior social interaction showed stable Drug-seeking-responsive DAergic activity across sessions, ruling out extinction effects (Supplementary Fig. 7p–r) and indicating that the observed reduction in Drug-seeking-Ens[DA] requires prior social reward experience.

Given that drug pursuit often occurs at the expense of natural rewards, particularly social reward[24], we investigated the reciprocal possibility: whether Drug-seeking-Ens[DA] inhibits Social-reward-Ens[DA]. Using a similar dual-labeling strategy to selectively target Drug-seeking-Ens[DA] (Label 1) and Social-reward-Ens[DA] (Label 2) (Supplementary Fig. 12a–d), optogenetic stimulation of Drug-seeking-Ens[DA] elicited robust outward currents in ~78% of tested Social-reward-Ens[DA] ($n = 4$ animals, 18/23 neurons; Supplementary Fig. 12e). These currents were abolished by TTX and rescued by co-application of TTX and 4-AP (Supplementary Fig. 12e), indicating monosynaptic connectivity. Furthermore, the currents were also eliminated by PTX ($n = 3$ animals, 11 neurons; Supplementary Fig. 12f), but were unaffected by NBQX ($n = 3$ animals, 10 neurons; Supplementary Fig. 12g) or SCH-23390 ($n = 3$ animals, 10 neurons; Supplementary Fig. 12h), confirming that these currents represent GABA_A receptor-mediated IPSCs. Consistently, photostimulation of Drug-seeking-Ens[DA] terminals produced a sustained reduction in the firing rate of ~62% of tested Social-reward-Ens[DA] ($n = 3$ animals, 8/13 neurons; Supplementary Fig. 12i), supporting a net inhibitory influence.

To assess whether cocaine seeking experience alters Social-reward-Ens[DA] responses, we longitudinally tracked these neurons under varying drug-seeking conditions (cocaine relapse test vs. no relapse test; $n = 3$ animals, 319 neurons; Supplementary Fig. 12j,k). Following a 1-h drug seeking during the cocaine relapse test, the proportion of DAergic neurons responsive to social stimuli was significantly reduced, although response magnitude remained unchanged (Supplementary Fig. 12l–n). This reduction was not due to social familiarity, as Social-reward-Ens[DA] responses remained stable across repeated social interactions without cocaine seeking (Supplementary Fig. 8p–r), suggesting that prior drug-seeking experience is required. Together, these findings support a model of reciprocal inhibition, wherein Social-reward-Ens[DA] and Drug-seeking-Ens[DA] dynamically compete to shape behavioral output.

## Social-reward-Ens[DA] mimics the protective effect of social reward

Next, we tested whether activation of Social-reward-Ens[DA] is sufficient and/or necessary to prevent drug seeking during the relapse test. We bilaterally injected AAV-cfos-ERCreER-PEST and AAV-TH-DIO-hM3Dq-mCherry into the VTA to label Social-reward-Ens[DA], and assessed whether chemogenetic reactivation via clozapine-N-oxide (CNO) attenuates cue-induced cocaine seeking (Fig. 4a, b). Electrophysiological recordings confirmed virus efficacy, with CNO increasing spontaneous firing of Social-reward-Ens[DA] (Fig. 4c). Behaviorally, CNO-mediated Social-reward-Ens[DA] activation reduced cue-induced active nose-pokes during the cocaine relapse test, without affecting inactive nose-pokes (Fig. 4d–f), indicating that Social-reward-Ens[DA] reactivation mimics the therapeutic effect of social reward against drug seeking. Optogenetic activation of Social-reward-Ens[DA] further validated this observation (Supplementary Fig. 13a–f). Additionally, control experiments activating random VTA DAergic neurons using AAV-hSyn-Cre and AAV-TH-DIO-hM3Dq-mCherry, following the methodology of a previously published study[25], showed no impact on drug seeking (Supplementary Fig. 13g–l), confirming the specificity of Social-reward-Ens[DA].

Given that activation of Social-reward-Ens[DA] attenuated drug seeking, we hypothesized that inhibiting these neurons would negate the protective effects of social reward. To test this, rats received bilateral injections of AAV-cfos-ERCreER-PEST and AAV-TH-DIO-hM4Di-mCherry into the VTA to label and silence Social-reward-Ens[DA]. CNO was administered 30 min prior to social interaction (Fig. 4g, h). Electrophysiological recordings confirmed reduced spontaneous firing of Social-reward-Ens[DA] following CNO treatment (Fig. 4i). The behavioral data showed that silencing Social-reward-Ens[DA] during social interaction increased cue-induced active nose-pokes, without affecting inactive nose-pokes (Fig. 4j–l), suggesting that the inhibition of Social-reward-Ens[DA] activity undermined the therapeutic effects of social reward against drug seeking during the relapse test.

## Social-reward-Ens[DA] and Drug-seeking-Ens[DA] exhibit distinct inputs

Rabies virus-based ensemble-specific monosynaptic tracing enables selective labeling of monosynaptic inputs to defined neuronal ensembles[26]. To compare the whole brain map of the inputs to Social-reward-Ens[DA] and Drug-seeking-Ens[DA], rats were infected with AAV-

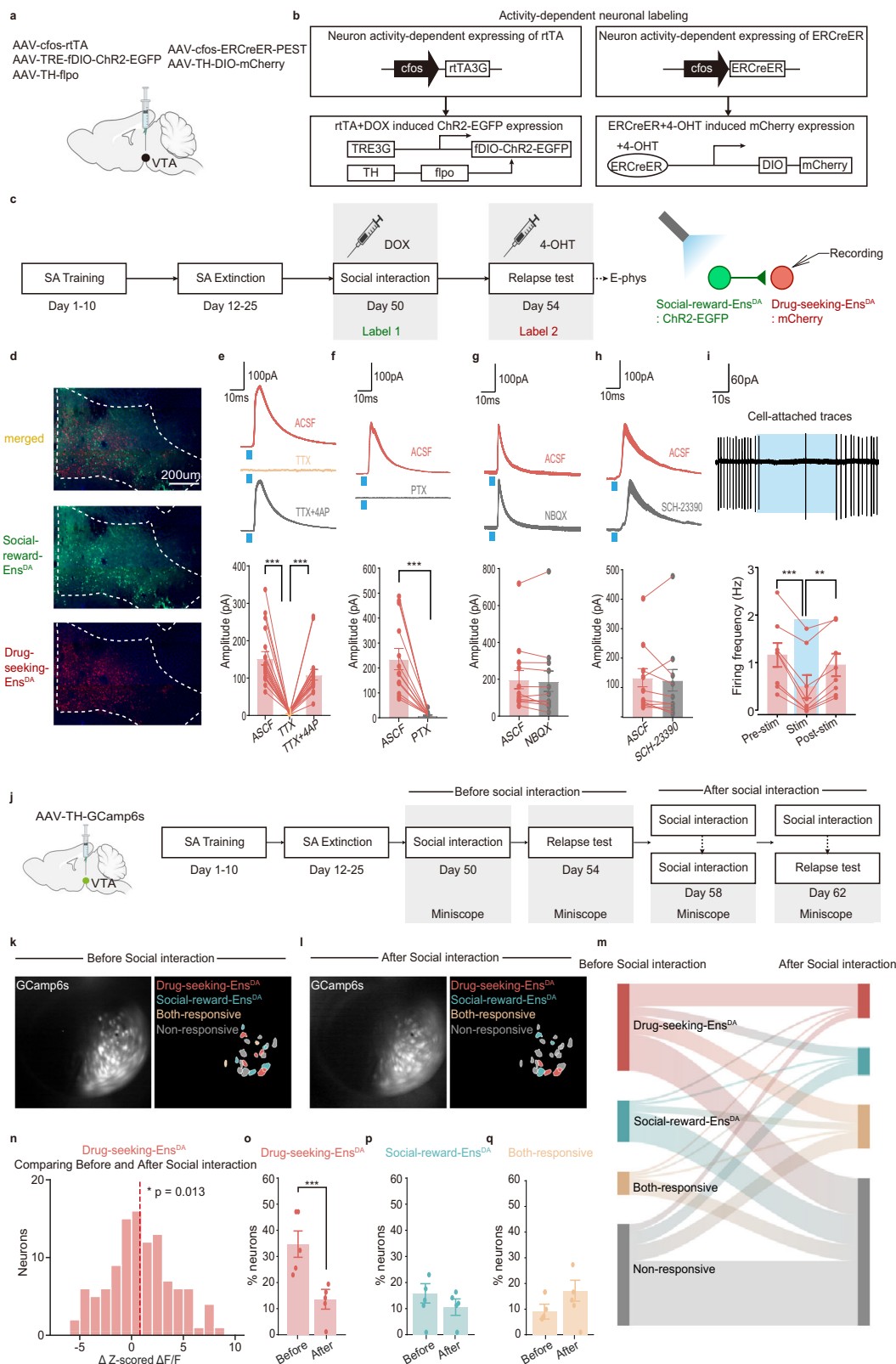

cfos-ERCreER-PEST and Cre-dependent helper viruses (AAV-TH-DIO-TVA-mCherry and AAV-DIO-RVG) into the VTA. Three weeks after labeling Social-reward-Ens[DA] and Drug-seeking-Ens[DA], rats were infected with the modified rabies virus RV-EvnA-△G-EGFP in the VTA. One week after infection, brains were collected for fluorescence imaging to identify EGFP-labeled input neurons (Fig. 5a, b). Starter ensembles expressing mCherry were restricted to the VTA (Fig. 5c). Notably,

additional EGFP-positive neurons within the VTA, not co-labeled with mCherry, suggested local projections from Social-reward-Ens[DA] to Drug-seeking-Ens[DA] (Fig. 5c). The normalized input index, which represents the ratio of EGFP-positive neurons to all input neurons, showed that Social-reward-Ens[DA] receives more inputs from the ventral striatum (VS), ventral pallidum (VP) and dorsal raphe nucleus (DRN) compared to the Drug-seeking-Ens[DA], while Drug-seeking-Ens[DA]

**Fig. 3 | Social-reward-Ens^DA in the VTA provides inhibitory input to Drug-seeking-Ens^DA. a** Virus injection strategy. **b** Activity-dependent neuronal labeling strategy. **c** Electrophysiology recording procedure. **d** Representative histology confirming successful labeling (repeated 4 times independently with similar results). **e** Representative traces of IPSCs. Light pulses reliably evoked IPSCs (red), TTX completely blocked IPSCs (yellow), and subsequent application of 4-AP in the presence of TTX rescued IPSCs (gray), with amplitudes normalized ($n = 18$ neurons, one-way ANOVA with Bonferroni multiple comparisons test, $F_{(2,34)} = 35.29$, $p < 0.001$; ACSF vs. TTX, $p < 0.001$; TTX vs. TTX + 4AP, $p < 0.001$). **f**–**h** Evoked IPSCs following application of PTX, NBQX, or SCH-23390; amplitudes normalized (PTX: $n = 13$ neurons, paired two-tailed t test: $t_{(12)} = 5.435$, $p < 0.001$). **i** Representative cell-attached traces. Photo-stimulation of Social-reward-Ens^DA led to a consistent decrease in the spontaneous firing of Drug-seeking-Ens^DA ($n = 8$ neurons, one-way

ANOVA with Bonferroni multiple comparisons test: $F_{(2,14)} = 26.48$, $p < 0.001$; Pre vs. Sti, $p < 0.001$; Sti vs. Post, $p = 0.007$). **j** Schematic of the experimental procedure for comparison of cell imaging before and after social interaction. **k**, **l** Example field of view across distinct imaging sessions. **m** Changes in neural tuning before versus after social interaction ($n = 110$ neurons, 5 male rats). **n** Changes of magnitude in response to drug-seeking after-before social reward experience ($n = 110$ neurons, two-tailed one-sample t-test, $p = 0.013$). **o**–**q** Changes of percentage of neurons responsive to drug seeking, social reward, or both ($n = 110$ neurons, chi-square test for two proportions, Drug-seeking-Ens^DA, $p < 0.001$; Social-reward-Ens^DA, $p = 0.239$; Both, $p = 0.073$; the error bars/bands show mean ± SEM). All the data are presented as mean ± SEM. 95% confidence interval was used for all statistical analyses in this figure. **a**, **c**, **j**: Created in BioRender. Xiaoxing, L. (2026) https://BioRender.com/hqs0eg6. $^{*}p < 0.05$, $^{**}p < 0.01$, $^{***}p < 0.001$.

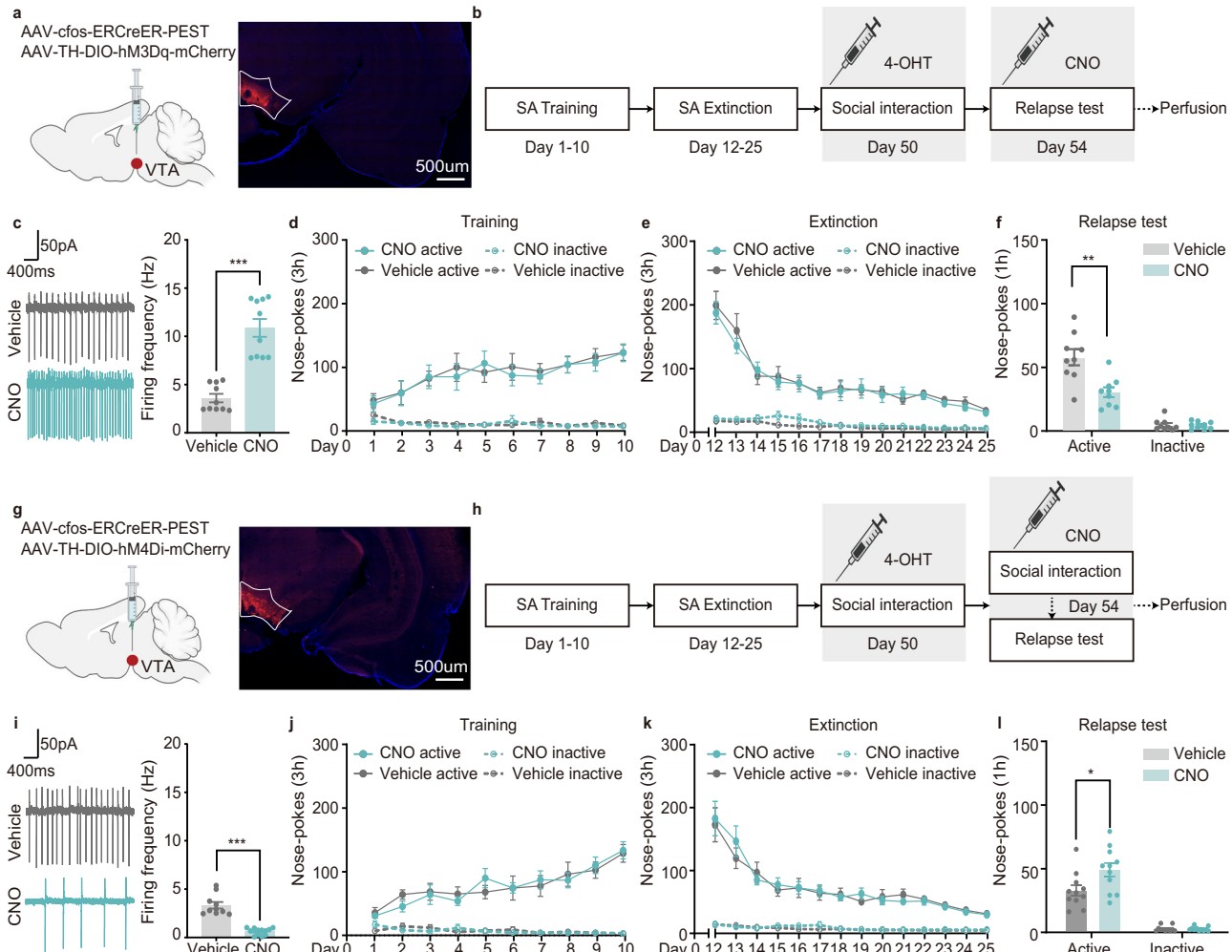

**Fig. 4 | Social-reward-Ens^DA modulates the protective effect of social reward against drug seeking. a** Virus injection strategy and representative hM3Dq-mCherry expression (scale bar: 500 μm; repeated 4 times independently with similar results). **b** Experimental scheme. **c** CNO increased spontaneous firing of hM3Dq-expressing Social-reward-Ens^DA ($n = 10$ neurons, unpaired two-tailed t test: $t_{(18)} = 7.102$, $p < 0.001$). **d** Number of nose-pokes during cocaine self-administration training ($n = 9$ male rats, three-way ANOVA, 10 consecutive days measurements from the same animal: $F_{(9,8)} = 0.274$, $p = 0.965$; see Supplementary Table 5A). **e** Number of nose-pokes during extinction ($n = 9$ male rats, three-way ANOVA, 14 consecutive days measurements from the same animal: $F_{(13,4)} = 0.534$, $p = 0.825$; see Supplementary Table 5B). **f** Number of nose-pokes during relapse test ($n = 9$ male rats, two-way ANOVA: $F_{(1,16)} = 12.222$, $p = 0.003$; see Supplementary Table 5C). **g** Virus injection strategy and representative hM4Di-mCherry expression (scale bar:

500 μm; repeated 4 times independently with similar results). **h** Experimental scheme. **i** CNO decreased spontaneous firing of hM4Di-expressing Social-reward-Ens^DA ($n = 10$ neurons, unpaired two-tailed t test: $t_{(18)} = 7.526$, $p < 0.001$). **j** Number of nose-pokes during cocaine self-administration training ($n = 11$ male rats, three-way ANOVA, 10 consecutive days measurements from the same animal: $F_{(9,12)} = 0.464$, $p = 0.873$; see Supplementary Table 5D). **k** Number of nose-pokes during extinction ($n = 11$ male rats, three-way ANOVA, 14 consecutive days measurements from the same animal: $F_{(13,8)} = 0.988$, $p = 0.528$; see Supplementary Table 5E). **l** Number of nose-pokes during relapse test ($n = 11$ male rats, two-way ANOVA: $F_{(1,20)} = 6.642$, $p = 0.018$; see Supplementary Table 5F). All the data are presented as mean ± SEM. 95% confidence interval was used for all statistical analyses in this figure. **a**, **b**, **g**, **h**: Created in BioRender. Xiaoxing, L. (2026) https://BioRender.com/hqs0eg6. $^{*}p < 0.05$, $^{**}p < 0.01$, $^{***}p < 0.001$.

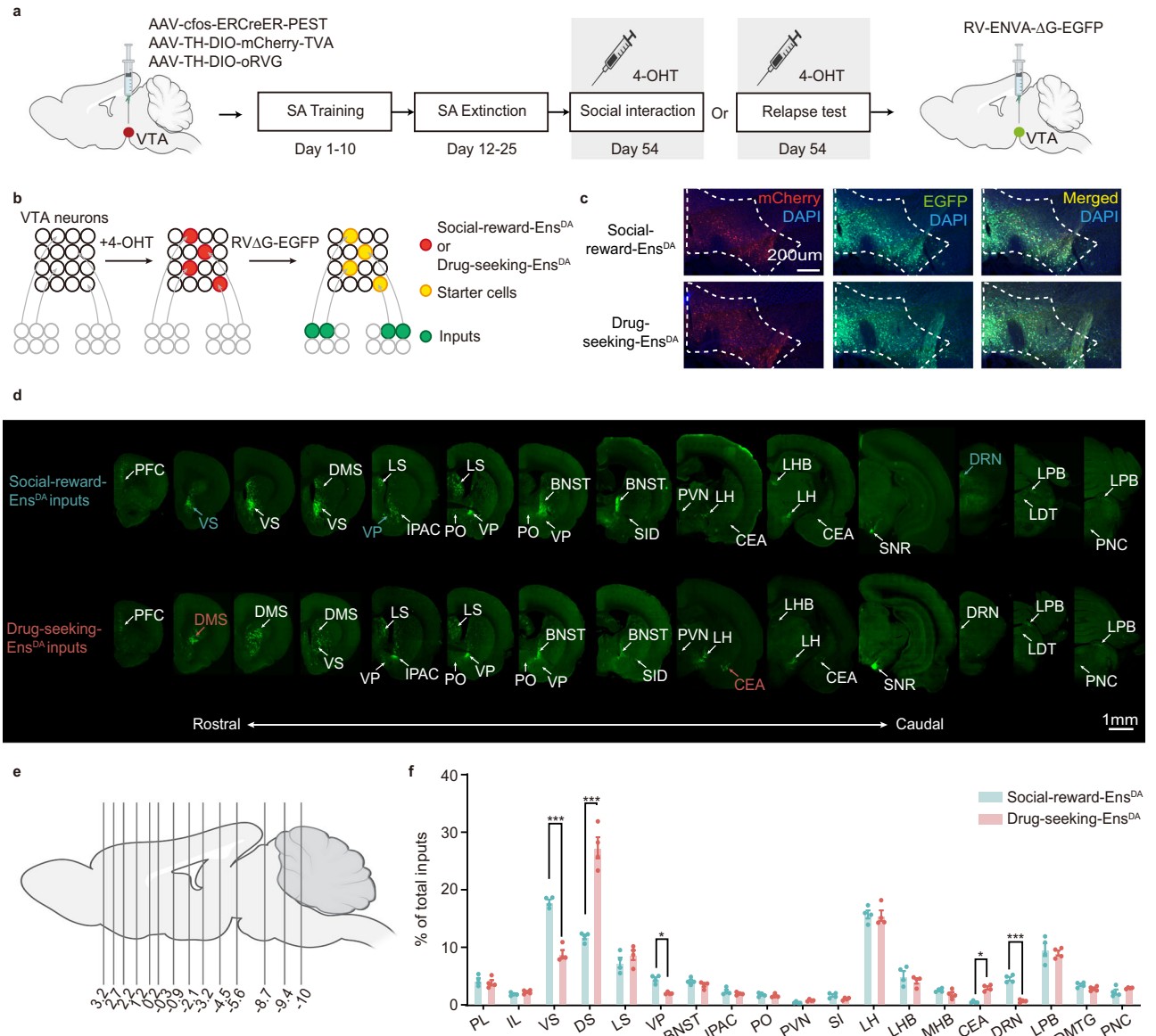

**Fig. 5 | Social-reward-Ens^DA and Drug-seeking-Ens^DA in the VTA establish different connection patterns. a, b** Schematic of the rabies-mediated monosynaptic retrograde labeling of the inputs onto VTA Social-reward-Ens^DA and Drug-seeking-Ens^DA. **c** Representative images of the VTA injection site and magnified view of starter cells with tamoxifen injection (repeated 4 times independently with similar results). Scale bar: 200 μm. **d** Representative images of the inputs (EGFP cells) from selected brain regions. Green: EGFP. **e** Series of coronal sections for monosynaptic inputs of VTA Social-reward-Ens^DA and Drug-seeking-Ens^DA. **f** Quantification of EGFP-positive neurons in inputs from each brain region, relative to the total number of EGFP-positive neurons (n = 4 male rats, two-way ANOVA with simple effect analysis, $F_{(18,57)} = 40.73$, $p < 0.001$; VS, $p < 0.001$; DS, $p < 0.001$; DRN, $p < 0.001$; VP, $p = 0.015$; CEA, $p = 0.016$). All the data are presented as mean ± SEM. 95% confidence interval was used for all statistical analyses in this figure. **a, e**: Created in BioRender. Xiaoxing, L. (2026) https://BioRender.com/hqs0eg6. *$p < 0.05$, **$p < 0.01$, ***$p < 0.001$.

receives more inputs from the dorsal medial striatum (DMS) and central amygdala (CeA) compared to the Social-reward-Ens^DA (Fig. 5d–f), which suggested that Social-reward-Ens^DA and Drug-seeking-Ens^DA establish different connections with brain regions.

## The DRN-VTA pathway regulates the therapeutic effect of social reward

Our findings that Social-reward-Ens^DA bidirectionally regulates drug seeking and preferentially receives input from the DRN, along with selective cfos activation in DRN-projecting but not VS- or VP-projecting VTA DAergic neurons during social interaction (Supplementary Fig. 14a–n), led to a clear hypothesis that activation of VTA DAergic neurons that receive input from the DRN would be sufficient to suppress drug seeking. To test this, we combined an anterograde

monosynaptic virus (AAV2/1-TH-flpo) with a flpo-dependent DREADD virus (AAV-FDIO-hM3Dq-EGFP) to selectively activate VTA DAergic neurons that receive input from the DRN (Fig. 6a, b and Supplementary Fig. 15a, b). Electrophysiological validation confirmed that CNO administration robustly increased the firing frequency of targeted VTA DAergic neurons (Fig. 6c and Supplementary Fig. 15c). Crucially, at the behavioral level, activation of VTA DAergic neurons that receive input from the DRN reduced cue-induced active nose-pokes in both the cocaine and heroin relapse tests, without influencing inactive nose-pokes (Fig. 6d–f and Supplementary Fig. 15d–f), suggesting that activating VTA DAergic neurons that receive input from the DRN mimic the therapeutic effect of social reward in preventing drug seeking. In contrast, parallel activation of VTA DAergic neurons that receive input from VP or VS using the same approach failed to alter drug seeking

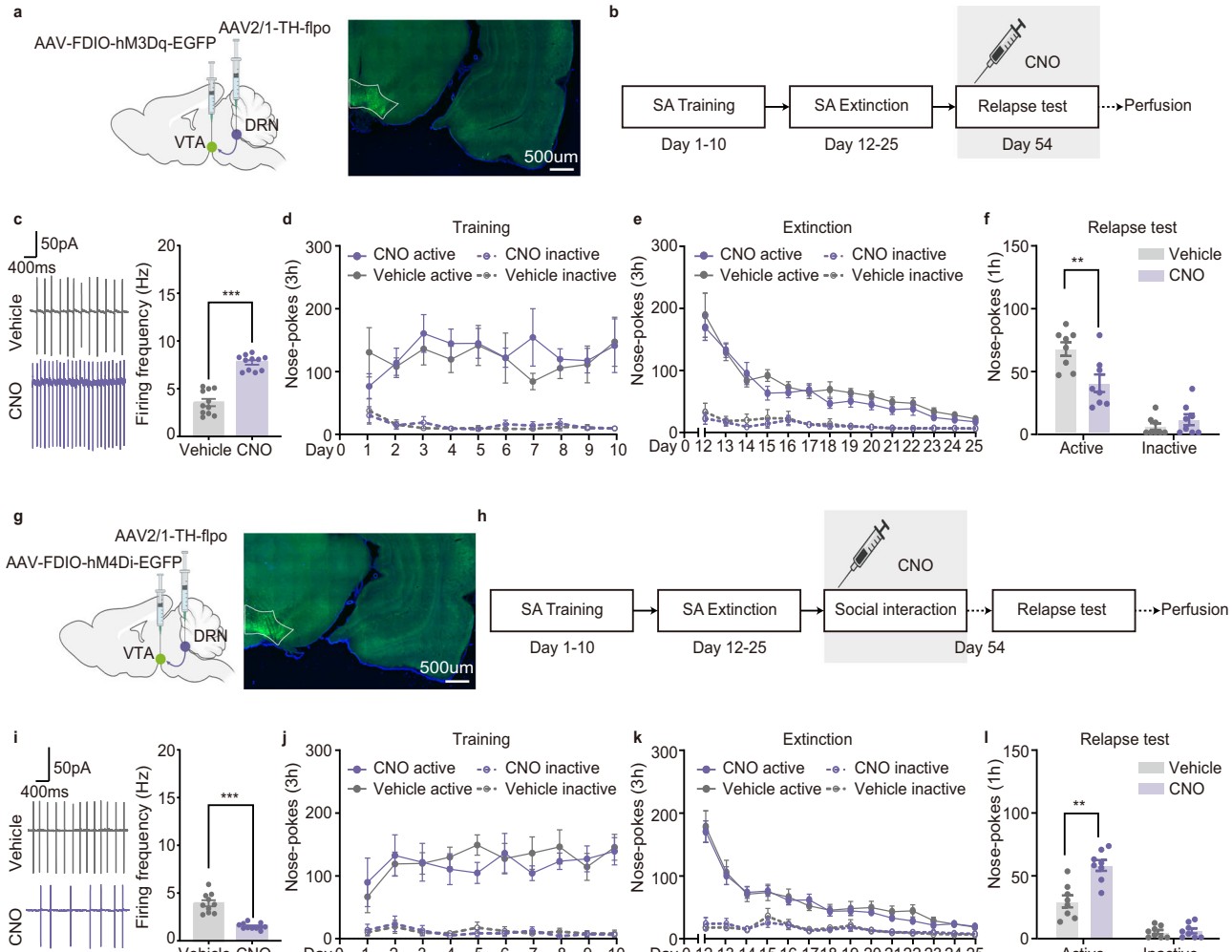

**Fig. 6 | VTA DAergic neurons that receive input from the DRN regulate the therapeutic effect of social reward against drug seeking. a** Virus injection strategy and representative hM3Dq-EGFP expression (scale bar: 500 µm; repeated 4 times independently with similar results). **b** Experimental scheme. **c** CNO increased firing of DRN-input DAergic neurons ($n = 11$ neurons, unpaired two-tailed t test: $t_{(20)} = 9.373, p < 0.001$). **d** Number of nose-pokes during cocaine self-administration training ($n = 8$ male rats, three-way ANOVA, 10 consecutive days measurements from the same animal: $F_{(9,6)} = 0.843, p = 0.607$; see Supplementary Table 8A). **e** Number of nose-pokes during extinction ($n = 8$ male rats, three-way ANOVA, 14 consecutive days measurements from the same animal: $F_{(13,2)} = 6.508, p = 0.141$; see Supplementary Table 8B). **f** Number of nose-pokes during relapse test ($n = 8$ male rats, two-way ANOVA: $F_{(1,14)} = 8.673, p = 0.011$; see Supplementary Table 8C). **g** Virus injection strategy and representative hM4Di-EGFP expression (scale bar: 500 µm;

repeated 4 times independently with similar results). **h** Experimental scheme. **i** CNO decreased firing of DRN-input DAergic neurons ($n = 10$ neurons, unpaired two-tailed t test: $t_{(18)} = 6.991, p < 0.001$). **j** Number of nose-pokes during cocaine self-administration training ($n = 8$ male rats, three-way ANOVA, 10 consecutive days measurements from the same animal: $F_{(9,6)} = 1.439, p = 0.339$; see Supplementary Table 8D). **k** Number of nose-pokes during extinction ($n = 8$ male rats, three-way ANOVA, 14 consecutive days measurements from the same animal: $F_{(13,2)} = 1.143$, $p = 0.560$; see Supplementary Table 8E). **l** Number of nose-pokes during relapse test ($n = 8$ male rats, two-way ANOVA: $F_{(1,14)} = 15.180, p = 0.002$; see Supplementary Table 8F). All the data are presented as mean ± SEM. 95% confidence interval was used for all statistical analyses in this figure. **a**, **b**, **g**, **h**: Created in BioRender. Xiaoxing, L. (2026) https://BioRender.com/hqs0eg6. $^{*}p < 0.05$, $^{**}p < 0.01$, $^{***}p < 0.001$.

(Supplementary Fig. 16a–l), confirming the circuit specificity of the DRN input. To rule out the effect of potential retrograde transport of AAV2/1-Cre[27] and make our conclusion more convincing, we established control groups to dissect the functional roles of DRN-VTA projections versus VTA-DRN projections. The results showed that activation of the DRN-VTA serotonergic projections, rather than VTA-DRN dopaminergic projections, mimicked the therapeutic effect of social reward against drug seeking (Supplementary Fig. 17a–j), confirming the critical role of DRN serotonergic inputs.

To further investigate the functional relevance of VTA DAergic neurons that receive input from the DRN, we used the same virus strategy to inhibit this projection. AAV2/1-TH-flpo was injected into DRN, and AAV-fDIO-hM4Di-EGFP was injected into VTA (Fig. 6g, h and Supplementary Fig. 15g, h). Electrophysiological recordings confirmed that CNO administration reduced the spontaneous firing of EGFP-

labeled DAergic neurons (Fig. 6i and Supplementary Fig. 15i). At the behavioral level, inhibiting VTA DAergic neurons that receive input from the DRN during the social interaction significantly increased cue-induced active nose-pokes both in the cocaine and heroin relapse tests, without affecting inactive nose-pokes (Fig. 6j–l and Supplementary Fig. 15j–l), suggesting that inhibition of VTA DAergic neurons that receive input from the DRN could reverse the therapeutic effect of social reward against drug seeking during the relapse test.

## Discussion

While drugs of abuse hijack the brain's reward circuitry[2,3], whether natural rewards can counteract these effects has remained unresolved. In this study, we reveal that social reward and drug seeking engage distinct DAergic ensembles in the VTA, with antagonistic Social-reward-Ens[DA] and Drug-seeking-Ens[DA] balancing social reward and drug

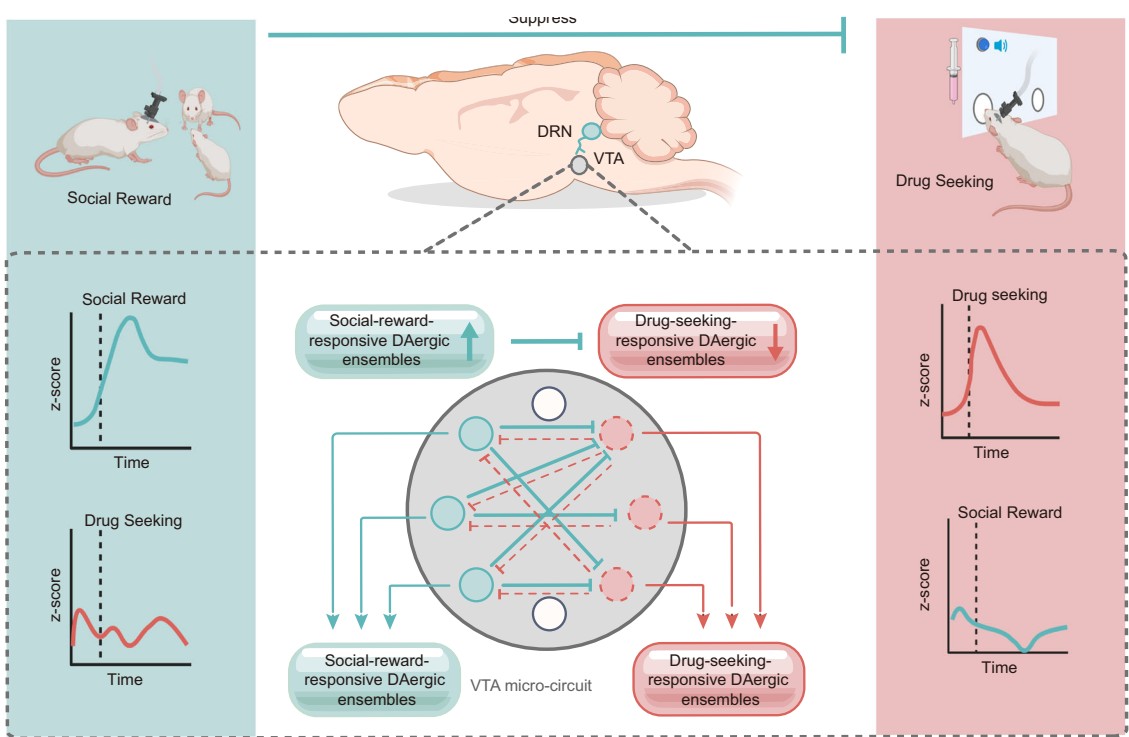

**Fig. 7 | A working model illustrating how social reward modulates drug seeking through dynamic, reciprocal inhibition between Social-reward-Ens[DA] and Drug-seeking-Ens[DA].** Social reward engages the DRN-VTA circuit to activate Social-reward-responsive DAergic ensembles, which suppress drug-seeking-responsive DAergic ensembles and attenuate relapse (Created in BioRender. Xiaoxing, L. (2026) https://BioRender.com/hqs0eg6).

seeking through dynamic competitive interactions. Furthermore, these ensembles exhibit divergent connectivity, activating DRN-VTA pathway can suppress drug seeking. These findings resolve a long-standing paradox by identifying a neural substrate through which social rewards mitigate drug seeking, offering a mechanistic framework for interventions that harness natural reward to combat SUD (Fig. 7).

The behavioral consequences of social interaction regulation in SUD hinge critically on the drug history of social partners (drug-naïve versus drug-exposed). In general, drug intake and expression can be enhanced by interaction with a drug-exposed partner but attenuated by engaging with a drug-naïve peer[6,28,29]. Consistent with these observations, our results demonstrate that interactions with drug-naïve partners suppress drug seeking during the relapse test. Notably, we found that the protective effect of acute social reward (with drug-naïve partners) is transient, disappearing after 24 h (Supplementary Fig. 18a, b). Consistently, fiber photometry recordings from the VTA and NAc revealed that drug-seeking-induced calcium signal and DA release elevations were attenuated immediately following social interaction, but this attenuation was no longer observed 24 h later (i.e., 4 days after the initial social interaction). This time-limited protective phenotype may parallel the contingency management in clinical addiction intervention[30]: in the absence of non-drug reward reinforcers, drug-seeking behavior is highly prone to relapse[31,32]. Furthermore, social housing environments (with drug-naïve partners), characterized by sustained conspecific interaction, exert a long-term protective influence on drug-seeking behavior. Specifically, such housing conditions accelerated the extinction of drug-associated memory and suppressed drug relapse (Supplementary Fig. 18c, d).

Accumulating evidence indicates that DAergic neurons may mediate the dual effects of social interaction on SUD[5]. A recent study demonstrated that corticotropin-releasing hormone release from the piriform cortex to the amygdala is selectively enhanced during negative social interaction (with drug-exposed partners)[33]. In contrast, our study showed that the DRN-VTA pathway is selectively enhanced during social reward (with drug-naïve partners); specifically, Drug-seeking-Ens[DA] receives greater input from the amygdala, whereas Social-reward-Ens[DA] receives greater input from the DRN. Thus, we hypothesize that the protective effect of social reward is mediated by activation of the DRN-VTA pathway, whereas the opposing effect of negative social interaction may arise from excessive activation of the amygdala-VTA pathway.

DAergic neurons in the VTA are implicated in diverse processes, including emotion[34,35], memory[36,37], motivation[38], and SUD[39,40]. This functional diversity arises from the VTA's heterogeneity. Its DA sub-populations differ in transcriptional identity, anatomical connectivity, and behavioral engagement[41,42]. Recent transcriptomic studies classify DAergic neurons into 15 distinct subtypes, each showing unique responses to task variables during complex behaviors[13,43]. Notably, specific subpopulations selectively encode positive or negative motivational signals[14,44], suggesting that discrete DA ensembles may process distinct stimuli, such as natural reward versus drug seeking. However, a recent study revealed a significant overlap in MSNs of NAc responding to natural rewards (drinking/feeding) and addictive substances[2], suggesting that overlapping DAergic ensembles may encode these two sets of stimuli. Thus, whether the same or different DAergic subpopulations encode natural rewards and drug-related behaviors remains unclear. Here, we demonstrate that social reward and drug seeking engage minimally overlapping DAergic ensembles, and they establish different connections with brain regions. These findings are consistent with previous studies and further support the unique role of distinct DAergic population[44,45].

While distinct DAergic subpopulations in the VTA have been well-characterized, their functional interactions remain unclear. Here, we demonstrate that selective activation of Social-reward-Ens[DA] suppresses drug seeking, implicating this population in the modulation of

drug-seeking behaviors. Notably, midbrain DA neurons are known to co-release GABA alongside DA[22,46]. Building on this evidence, we propose that Social-reward-Ens[DA] may exert inhibitory control over Drug-seeking-Ens[DA]. Consistent with this hypothesis, we found that optogenetic stimulation of Social-reward-Ens[DA] evoked robust, picrotoxin-sensitive IPSCs in Drug-seeking-Ens[DA], confirming GABAergic transmission. Conversely, stimulation of Drug-seeking-Ens[DA] also evoked IPSCs in Social-reward-Ens[DA]. These results identify a bidirectional inhibitory microcircuit between Social-reward-Ens[DA] and Drug-seeking-Ens[DA] within the VTA. We speculate that this reciprocal interaction is progressively established during repeated drug use, although we cannot fully exclude the existence of genetically predetermined cell types in the VTA; further investigation will be needed to dissect these possibilities.

The DRN has been classically recognized as a nucleus that contains an anatomically and functionally diverse population of serotonergic neurons[47], many of which innervate brain areas involved in reward processing[48]. Social interaction, a behavior with high reward salience, has been linked to serotonergic activity in the DRN[49]—a link our findings corroborate and extend. Brain-wide monosynaptic mapping revealed that Social-reward-Ens[DA] receives substantial inputs from DRN, and social interaction robustly activates DRN populations that project to the VTA[50]. Prior work demonstrates that serotonergic DRN-VTA projections enhance social interaction time in stress-susceptible mice, while silencing this pathway diminishes social interaction time[51]. Consistent with this, we found that activation of VTA DAergic neurons that receive input from the DRN or the serotonergic DRN-VTA projections mimicked the therapeutic effect of social reward against drug seeking (Supplementary Fig. 17a–e). These findings provide partial mechanistic support for models positing serotonin's role in mitigating compulsive drug-seeking[52,53].

Beyond the inputs to DAergic neurons, understanding how their downstream outputs drive opposing behaviors remains a critical question in the field[54,55]. A recent study has highlighted the role of distinct DAergic downstream projections in mediating the influence of social interaction on addiction susceptibility: optogenetic activation of the DAergic pathway from the VTA to prefrontal cortex suppressed drug seeking, whereas activation of the VTA-to-nucleus accumbens (NAc) pathway exerted the opposite effect[5]. Moreover, distinct VTA DAergic projections into defined NAc subregions also mediate diverse behavioral functions[55]. Building on these findings, investigating how distinct downstream dopamine signals functionally connect to the Social-reward-Ens[DA] and Drug-seeking-Ens[DA] identified in our study will be a key focus of future research.

In addition to social rewards, other natural rewards, such as palatable food and physical exercise, have also been shown to mitigate the effects of drug use. For example, voluntary abstinence based on the choice between drugs and palatable food significantly reduces the incubation of drug-seeking behavior in rats[56]. Similarly, aerobic exercise has been demonstrated to attenuate cocaine-seeking behavior[32]. In line with these findings, our data suggest that sucrose reward can suppress cocaine seeking during the relapse test, albeit with a therapeutic efficacy lower than that of social reward (Supplementary Fig. 19a–d). Moreover, we found that activation of Drug-seeking-Ens[DA] significantly inhibited feeding behavior (Supplementary Fig. 19e–h), further supporting the competitive functional relationship between natural rewards and drug-seeking behavior.

In summary, we identify two functionally distinct DAergic ensembles that competitively process social reward and drug seeking. We establish that the Social-reward-Ens[DA] directly suppresses the Drug-seeking-Ens[DA] via GABAergic co-release, a previously unappreciated mechanism of neuronal competition. Furthermore, these ensembles are defined by divergent connectivity, with the Social-reward-Ens[DA] receiving preferential innervation by the DRN. These results challenge the traditional view of a monolithic reward system and provide a specific circuit-based framework for harnessing social reward to combat SUD.

## Methods

### Animals
Male Sprague-Dawley rats (280–300 g; Beijing Vital River, China) were group-housed (5/cage) for one week, then singly housed post-surgery. Rats were kept on a 12 h light/dark cycle (lights on at 20:00) with ad libitum food and water. Behavioral tests were conducted during the dark phase. Only male rats were used in the present study, and sex was not included as a variable in the study design or statistical analysis. This choice was made to minimize potential confounding effects related to sex hormones and estrous cycle fluctuations, which could introduce additional variability in the behavioral and physiological measures under investigation. All procedures followed the guidelines set forth by the Regulation for the Administration of Affairs Concerning Experimental Animals (China, 1988) and the National Institutes of Health Guide for the Care and Use of Laboratory Animals, with approval by the Biomedical Ethics Committee for Animal Use and Protection of Peking University (LA2021168).

### Jugular vein catheterization
Rats were anesthetized with isoflurane (4–5% induction, 1.5–2% maintenance). A silicone catheter (OD 1.0 mm, ID 0.55 mm) was inserted into the jugular vein and secured near the right atrium. After a 5-day recovery, catheters were flushed daily with 0.4 mL heparinized saline (100 U/mL) and penicillin (40,000 U/mL) until the end of cocaine self-administration training.

### Behaviour
**Self-administration operant apparatus.** Cocaine self-administration, extinction, and relapse tests were conducted in 20 sound-attenuated operant chambers (AniLab, Ningbo, China), each equipped with a house light, two nose-poke holes with infrared sensors, and a buzzer for auditory cues. Catheters were connected to pump-driven syringes. Data were recorded via AniLab software.

**Cocaine/heroin self-administration training.** Based on our previous studies[17,57], rats were trained to self-administer intravenous cocaine (0.75 mg/kg/inf) or heroin (0.05 mg/kg/inf) in 3-h daily sessions for 10 days under a fixed ratio-1 schedule. Sessions began with the house light on. An active nose-poke triggered a 5-s tone-light cue (CS), followed by drug infusion and a 20-s timeout (house light off), during which responses were recorded but not reinforced. Inactive nose-pokes had no consequence. To prevent overdose, infusions were capped at 21 per hour (max 63/session).

### Extinction training
Under identical conditions (3 h/day), extinction sessions omitted drug delivery while maintaining CS presentation after active nose-pokes. Training continued until active responses dropped to <20% of the average from the last 3 days of self-administration, sustained for 2 consecutive days.

### Relapse test
Four weeks after extinction, rats underwent a 1-h relapse test. As in extinction, active nose-pokes triggered the CS without drug delivery.

### Social reward treatment
Social reward treatment was conducted based on a previous study[15,58], two juvenile male rats (4 weeks old, 200–220 g) were placed in a box lined with soft bedding to acclimate to the new environment for 5–10 min. The test rat was then introduced into the box for a 30-min period, during which the test rat and juvenile rats were allowed to freely interact.

**No reward treatment (exposure box control)**

Rats were placed alone in the same box for 30 min, without any conspecifics.

**Toy rat interaction treatment (novelty control)**

Two lifelike toy rats were placed in the box, and the test rat was allowed to interact freely for 30 min.

**Anesthetized rat interaction treatment (novelty control)**

Two juvenile male anesthetized rats (4 weeks old, 200–220 g) were placed in the same box. The test rat was then introduced and allowed to interact freely with the anesthetized rats for 30 min.

**Sucrose water reward treatment (other natural reward control)**

Adapted from a previous study[2], rats were water-deprived for 16–24 h to induce a thirst state. After 5–10 min of habituation in the box, they were given free access to sucrose water for 1 h.

**Social conditioned place preference**

The social conditioned place preference (CPP) procedure was performed based on a previous study[16]. On Day 1, rats were placed in the center of a three-chamber CPP apparatus and allowed to explore freely for 15 min to assess baseline preferences. Animals showing strong initial side preference were excluded. On Day 2, rats underwent 24 h of social conditioning (with two juvenile males) in one chamber and 24 h of isolation in the opposite chamber. Chamber assignments were counterbalanced. On Day 4, a 15-min post-conditioning test was conducted. The CPP score was calculated as the time spent in the social-paired chamber minus that in the isolate-paired chamber.

**Social self-administration**

A social self-administration paradigm was conducted in sound-attenuated operant chambers (AniLab, Ningbo, China) for 8 days. Rats were trained to self-administer to gain access to social targets (two juvenile conspecifics) during daily 60-min sessions. An active nose poke triggered the automated gate to open for 60 s, allowing the rat access to the social targets in a smaller chamber. The social target chamber had an open side covered with holes spaced 1 cm apart, enabling the rats to freely interact, smell, and investigate each other without either animal entering the other chamber. An inactive nose poke, located opposite the active nose poke, had no consequences. Data were recorded via AniLab software.

**Optical intracranial self-stimulation (oICSS)**

To investigate whether interaction with juvenile conspecifics is rewarding, optogenetic techniques were employed to specifically activate social reward-Ens. oICSS training was conducted in operant chambers (AniLab, Ningbo, China) equipped with two nose pokes (active and inactive). Two weeks following stereotaxic surgery, the animals were placed in the chambers for daily behavioral training. A laser generator was triggered via a Master-9 pulse stimulator (Thinkerbiotech, Nanjing, China). During each 60-min oICSS training session, an active nose poke elicited a 5-s tone and laser stimulation (1 s, 10 mW, 40 Hz, 5 ms pulse width); an inactive nose poke had no consequences. The number of active and inactive nose pokes was recorded per session. Starting on day 5 of testing, the active nose poke delivering laser stimulation was switched to the opposite location (previously the inactive nose poke) and maintained until testing concluded on day 8. Scheduling of experimental events and data collection was performed using AniLab software.

**Adeno-associated virus (AAV) injection**

Procedures followed our previous protocol[59], Rats were anesthetized with isoflurane (4–5% induction; 1.5–2% maintenance) and placed in a stereotaxic frame. AAVs were injected at 30 nL/minute using a glass pipette connected to a microsyringe via mineral oil-filled tubing. The pipette was left in place for ≥10 min post-injection to facilitate diffusion before slow withdrawal.

Calcium imaging in VTA: AAV-TH-GCamp6s (AAV2/9, ≥2 × 10^12 vg/mL, 300 nL) was unilaterally injected into the VTA.

DA release detection in NAc: AAV-hSyn-GRAB-DA3.3 (AAV2/9, 5 × 10^12 vg/mL, 300 nL) was injected into the NAc.

DRN-VTA imaging: AAV-TH-Cre (AAV2/R, ≥5 × 10^12 vg/mL, 250 nL) was injected into VTA; AAV-DIO-GCamp6s (AAV2/9, 5 × 10^12 vg/mL, 250 nL) into DRN.

IEG labeling of Social-reward-Ens^DA and Drug-seeking-Ens^DA: AAV-cfos-ERCreER (AAV2/9, 5 × 10^11 vg/mL) and AAV-TH-DIO-mCherry (AAV2/8, 2 × 10^13 vg/mL) were mixed 1:1 and injected into VTA (250 nL).

Chemogenetic manipulation of Social-reward-Ens^DA: Activation: AAV-cfos-ERCreER (AAV2/9, 5 × 10^11 vg/mL) + AAV-TH-DIO-hM3Dq-mCherry (AAV2/8, 2 × 10^13 vg/mL); Inhibition: AAV-cfos-ERCreER (AAV2/9, 5 × 10^11 vg/mL) + AAV-TH-DIO-hM4Di-mCherry (AAV2/8, 2 × 10^13 vg/mL); All mixed 1:1 (250 nL/side).

Optogenetic activation of Social-reward-Ens^DA: AAV-cfos-ERCreER (AAV2/9, 5 × 10^11 vg/mL) + AAV-TH-DIO-ChR2-mCherry (AAV2/8, 2 × 10^13 vg/mL); All mixed 1:1 (250 nL/side).

Monosynaptic input tracing: AAV-cfos-ERCreER (AAV2/9, 5 × 10^11 vg/mL), AAV-TH-DIO-TVA-mCherry (AAV2/8, 2 × 10^13 vg/mL), and AAV-TH-DIO-RVG (AAV2/8, 2 × 10^13 vg/mL) were mixed 1:1:1 and injected into VTA (250 nL). Three weeks later, RV-EnvA-ΔG-EGFP (≥2 × 10^8 IFU/mL, 250 nL) was injected at the same site.

Electrophysiological recording: AAV-cfos-rtTA (AAV2/9, 5 × 10^12 vg/mL), AAV-TH-Flp (AAV2/8, 5 × 10^12 vg/mL), and AAV-TRE-fDIO-ChR2-EGFP (AAV2/8, 5 × 10^12 vg/mL) were mixed 1:1:1 and injected into VTA (250 nL). Two weeks later, AAV-cfos-ERCreER (AAV2/9, 5 × 10^11 vg/mL) and AAV-TH-DIO-mCherry (AAV2/8, 2 × 10^13 vg/mL) were injected into VTA (1:1, 250 nL).

VTA DAergic neurons that receive input from the DRN manipulation: Activation: AAV-TH-Flp (AAV2/1, 5 × 10^13 vg/mL, 300 nL) into DRN (AP − 7.0, ML ± 0.0, DV − 6.0 mm); AAV-TRE-fDIO-hM3Dq-EGFP (AAV2/9, 5 × 10^12 vg/mL, 250 nL) into VTA. Inhibition: same coordinates; AAV-TRE-fDIO-hM4Di-EGFP (AAV2/9) into VTA.

DRN^SERT-VTA manipulation: AAV-SERT-Cre (AAV2/9, ≥2 × 10^12 vg/mL, 250 nL) into DRN; AAV-DIO-hM3Dq-mCherry (AAV2/9, 5 × 10^12 vg/mL, 250 nL) into VTA.

**Activity-dependent neuronal labeling**

Activity-dependent neuronal tagging followed previous studies[60–62], with suitable modifications. 4-hydroxytamoxifen (4-OHT; Sigma, H6278) was dissolved at 20 mg/mL in ethanol (37 °C, 15 min), then diluted with corn oil (Sigma, C8267) to 10 mg/mL. Ethanol was removed by vacuum centrifugation. 4-OHT solution was stored at 4 °C for up to 24 h before use. Doxycycline (Dox; Sigma, D9891) was dissolved in saline at 5 mg/mL. Both were administered via intraperitoneal injection.

To minimize immediate-early gene activation from transport stress, rats were moved to an adjacent room ≥3 h before injection. Social-reward-responsive DAergic ensembles were tagged by 4-OHT (50 mg/kg, i.p.) 30 min before the social interaction test or Dox (50 mg/kg, i.p.) 1 h before the social interaction test. Drug-seeking ensembles were tagged by 4-OHT (50 mg/kg, i.p.) 30 min before the relapse test.

**Chemogenetic activation or silencing**

Procedures followed previous studies[61,63]. To activate the activity of Social-reward-Ens^DA, at least two weeks subsequent to the virus injection surgery, the rats underwent an intraperitoneal injection of 4-OHT 30 min prior to the social interaction. Roughly another 4 days later, the same group of rats received an intraperitoneal injection of clozapine N-oxide hydrochloride (CNO, 5 mg/kg, dissolved in saline; MCE, HY-

17366) 30 min before the drug relapse test. To silence the activity of Social-reward-Ens[DA], at least two weeks after the virus injection surgery, the rats were intraperitoneally injected with 4-OHT 30 min before the social interaction. Approximately another 4 days later, the same rats were intraperitoneally injected with CNO 30 min before the social interaction, and then they immediately took a drug relapse test.

To activate the activity of VTA DAergic neurons that receive input from the DRN, at least 2 weeks after the virus injection surgery, rats received an intraperitoneal injection of CNO 30 min before the drug relapse test. To silence the activity of VTA DAergic neurons that receive input from the DRN, at least two weeks after the virus injection surgery, rats were intraperitoneally injected with CNO 30 min before the social interaction, and then they immediately took a drug relapse test.

## Optogenetic activation
The experimental procedures were performed as described in a previous study[32]. To activate Social-reward-Ens[DA], at least two weeks after virus injection surgery, fiber optic probes were implanted at stereotaxic coordinates targeting the VTA (AP −5.9, ML ± 0.5, DV −8.1 mm). Following a recovery period after probe implantation, the rats received an intraperitoneal injection of 4-OHT 30 min prior to the social interaction session on day 50. Approximately 4 days later, the same rats underwent photo-stimulation (473 nm, 10-ms pulse duration, 20 Hz, 10 mW) delivered in 3 cycles: 5 min on followed by 10 min off for the first two, with the final cycle consisting of 5 min of stimulation immediately followed by the cocaine relapse test.

## Fiber implantation surgery for calcium imaging
Rats were anesthetized with isoflurane (4–5% induction; 1.5–2% maintenance) and secured in a stereotaxic frame. The scalp was shaved, disinfected, and a midline incision was made to expose the skull. Fiber optic probes were implanted at stereotaxic coordinates targeting the VTA (AP −5.9, ML ± 0.5, DV −8.1 mm) and NAc (AP + 2.1, ML ± 1.4, DV −6.6 mm). Probes were lowered through craniotomies to the target depth and fixed with dental cement. The scalp was sutured in layers. After recovery, calcium imaging was performed during social interaction, relapse, or stimulus exposure.

## GRIN lens implantation for single-neuron calcium imaging
Rats were anesthetized with isoflurane (4–5% induction; 1.5–2% maintenance) and placed in a stereotaxic frame. After scalp incision and periosteum removal, the skull was cleaned with sterile saline and thinned using a microsurgical drill. A small dural incision was made. A 1 mm diameter, 0.5 NA GRIN lens (CLHS100GFT130, Gofoton) was inserted above the VTA (AP −5.9, ML ± 0.5, DV −8.1 mm; ipsilateral to viral injection, referenced to bregma). The lens was lowered 0.8 mm and raised 0.4 mm to minimize tissue damage and set the final depth. It was secured with cyanoacrylate and sealed with dental cement.

After ~3 weeks of recovery, a baseplate was aligned and fixed with dental cement. Subsequent single-photon imaging sessions were conducted during behavioral tasks, including social interaction and drug relapse tests.

## Immunocytochemistry
Rats were deeply anesthetized with isoflurane (4–5%) and transcardially perfused with 4% paraformaldehyde. Brains were post-fixed for 24 h, cryoprotected in 20% and 30% sucrose in PBS (pH 7.2), and coronally sectioned at 40 μm.

Sections were washed (3 × 15 min in PBS), permeabilized with 0.1–0.5% Triton X-100 for 30 min at room temperature, and blocked with 5% BSA or serum in PBS for 1.5 h. Primary antibodies, mouse anti-cfos (1:1,000, Abcam, ab208942) or rabbit anti-tyrosine hydroxylase (1:2,000, Millipore, AB152), were applied overnight at 4 °C. After washing, slices were incubated for 2 h at room temperature with Alexa Fluor-conjugated secondary antibodies: goat anti-rabbit Alexa Fluor 647 (1:500, Invitrogen, A-21245) or donkey anti-mouse Alexa Fluor 488 (1:500, Invitrogen, A-32766).

Slices were counterstained with DAPI (Abcam, ab104139), mounted, and stored at 4 °C. Images were acquired with a fluorescence microscope (Olympus, 20× objective). Fluorescence-positive cells were manually quantified using ImageJ (v1.8.0).

## Electrophysiology
Cell-attached and whole-cell recordings were performed 2 weeks after labeling Social-reward-Ens[DA] and Drug-seeking-Ens[DA]. Following anesthesia, brains were rapidly extracted, and 300-μm horizontal VTA slices were prepared using a vibratome (Leica VT1200S) in ice-cold cutting solution (80 mM NaCl, 3 mM KCl, 26 mM NaHCO₃, 1.3 mM MgCl₂, 1 mM NaH₂PO₄, 20 mM glucose, 1 mM CaCl₂, 75 mM sucrose; bubbled with 95% $O_2$/5% $CO_2$). Slices were transferred to ACSF (124 mM NaCl, 3 mM KCl, 26 mM NaHCO₃, 1.3 mM MgCl₂, 1.5 mM CaCl₂, 1 mM NaH₂PO₄, 20 mM glucose; 95% $O_2$/5% $CO_2$) at 33 °C for 30 min and then maintained at room temperature.

Patch pipettes (2–4 MΩ, TW150F-3, WPI) were filled with Cs-based low Cl⁻ internal solution (110 mM Cs-methanesulfonate, 15 mM CsCl, 10 mM HEPES, 0.5 mM EGTA, 4 mM QX-314, 4 mM Mg-ATP, 0.3 mM GTP, 5 mM Na₂-phosphocreatine; pH 7.3 with CsOH, 270–280 mOsm). ChR2-expressing Social-reward-Ens[DA] neurons were optically stimulated via a 473 nm laser (1 ms pulses, 6.5–10 mW) through the back aperture of the objective. Light pulses were delivered at 30 s intervals.

VTA DAergic neurons were voltage-clamped at −70 mV (EPSCs) or 0 mV (IPSCs). Pharmacological agents (MCE unless noted): NBQX (10 μM), picrotoxin (10 μM), TTX (1 μM), 4-AP (0.1–1 mM) were bath-applied. Opto-evoked EPSCs/IPSCs were recorded for 1 min before and after drug application. Spontaneous activity was recorded in the cell-attached mode, with baseline, opto-stimulation (-1 min), and post-stim epochs.

Signals were acquired using a MultiClamp 700B amplifier (Axon Instruments), filtered at 2 kHz, digitized at 5 kHz (PCI-MIO-16E4, National Instruments), and recorded with pClamp 10 (Molecular Devices). Analysis methods followed those in our previous study[64]. Briefly, all electrophysiological data were analyzed using Mini Analysis software (Synaptosoft, USA). After automated detection, each trace was manually inspected to exclude miscounting, double selections, or noise-induced artifacts. Recordings were discarded if the series resistance (Rs) changed by more than 20% or exceeded 30 MΩ to ensure data stability and reliability.

## Data analysis for monosynaptic input
The method of data analysis for whole-brain tracing followed the approach outlined in previous literature[18]. The 19 regions chosen for analysis included the major inputs to the VTA that fell outside of the excluded region near the injection site. For VTA retro-transsynaptic tracing, GFP-positive input neurons were manually counted from every third 60-μm section throughout the entire brain, excluding the region near the VTA (5.2 mm to 6.0 mm from bregma). As the total number of input neurons varied among brains, we normalized the neuronal count for each area by the total number of input neurons counted in the respective brain. For each of the 19 brain regions, every third section was analyzed. If the total number of inputs in a region was fewer than five, every section was counted, and the counts were then multiplied by a scale factor. We did not adjust for the potential double-counting of cells in our quantifications, which likely led to slight overestimates, the extent of overestimation varied depending on the size of the cells within each quantified region.

## Fiber photometry recording and analysis
Fiber photometry recordings were conducted during social interaction and cocaine relapse tests using a 405/470-nm multi-channel system (Inper Technology, Hangzhou, China) to monitor calcium activity in

TH$^+$ VTA neurons and dopamine release in the NAc. Excitation light intensity at the fiber tip was maintained at 20–40 μW, and signals were sampled at 50 Hz.

Fluorescence data were analyzed using commercial software (Inper Technology, Hangzhou, China). To correct for motion artifacts and baseline drift, the 405 nm control signal was fitted via least-squares regression and subtracted from the 470 nm signal. $\Delta F/F$ was calculated as (470 signal–fitted 405 signal)/fitted 405 signal.

For relapse tests, signals were aligned to the onset of active nose pokes (first and last 10 s), while for social interaction, data were aligned to the onset of body contact. Significant calcium transients were defined as epochs in which $\Delta F/F$ exceeded 2.91 median absolute deviations (MADs) above baseline and remained at least 2 MADs above baseline for a sustained period. Event-related activity was quantified by computing the area under the curve (AUC), focusing on windows spanning 10 s before to 10 s after behavioral events. Traces were trial-aligned and averaged, with shaded error bands representing the SEM.

### Single-neuron calcium imaging recording and data analysis
**Calcium imaging acquisition.** A miniature fluorescence microscope system (Thinkerbiotech, Nanjing, China) was used to image calcium dynamics in freely moving rats. The system consisted of a compact lens, integrated optical and excitation modules (LED light source), and a camera for fluorescence acquisition. The excitation wavelength was selected based on the calcium indicator used. Acquisition parameters (e.g., frame rate, exposure time) were optimized to balance signal quality and minimize phototoxicity. During recording sessions, rats engaged in behavioral tasks such as nose-poking and social interaction.

The system was interfaced with an acquisition card via USB 3.0 and SMA coaxial cables, allowing synchronization with the operant behavior apparatus. The Sync output was connected to the behavioral control system to align neural activity with behavioral events. The microscope baseplate was affixed to the rat's skull using miniature screws. Imaging was performed after opening the main interface, loading the configuration file (User Config Example_V4_BNO_Miniscope.json), and verifying experimental parameters (e.g., file path, subject ID, LED power, frame rate, gain, and focal plane). The recording was initiated by clicking the "Run" and "Record" buttons.

**Calcium imaging pre-processing and fluorescence extraction.** Calcium imaging data were processed using commercial software (Thinkerbiotech, Nanjing, China) according to established procedures[10,65]. Raw videos (608 × 608 pixels, 30 Hz) were down-sampled to 304 × 304 pixels and 15 Hz. Motion correction was conducted using a template-matching plugin. Cellular fluorescence signals were extracted using the CNMFE algorithm in MATLAB (https://github.com/zhoupc/CNMF_E) and custom scripts (ROImap.m). ROIs were visually inspected to exclude non-cellular components based on morphological features.

### Cell registration across sessions
To identify the same neurons across experimental sessions, the Cell-Reg toolbox (https://github.com/zivlab/CellReg) and custom MATLAB code (aligned_cell_ activity.m) were used. Calcium traces were aligned to the onset of behavioral events (time 0 s). The [−2 s to −0 s] window served as the baseline, and cellular activity was z-scored across a [−10 s to +20 s] window.

### Definition of responsive neurons
Responsive neurons were identified using custom MATLAB code (responsive_neurons.m). Mean activity in the [0–20 s] post-event window across the first four trials was calculated. Neurons were classified as responsive to a given stimulus if their mean $z$-score exceeded 1.5.

### Overlap analysis
To assess whether social-reward- and drug-seeking-responsive DAergic subpopulations significantly overlapped, we compared the observed overlap rate to a null distribution. Responsive cell labels were randomly shuffled across all imaged neurons while maintaining the original number of cells responsive to each stimulus. For each of 10,000 permutations, the proportion of double-responsive neurons was computed. The empirical overlap rate was then compared to this null distribution. A significantly higher overlap than chance suggested a shared subpopulation activated by both stimuli. This analysis was conducted using custom MATLAB code (Comparing_the_overlap_fraction _with_shuffled_distribution.m).

### Statistical analysis
Following the experiments, all animals were euthanized for histological verification of viral expression and fiber placement. Only rats with correct injection and implant locations were included in the analyses. Behavioral data (e.g., number of infusions during cocaine/heroin self-administration and extinction) were analyzed using two-way ANOVA. A three-way ANOVA was used to assess active vs. inactive nose-pokes across sessions. For relapse data, a 2 × 2 repeated-measures ANOVA was performed. Significant ANOVA results were followed by simple effects analysis. All statistical data are presented as mean ± SEM. Analyses were conducted using GraphPad Prism v7.00 and IBM SPSS Statistics v24.00. Immunofluorescence quantification was performed in ImageJ. Statistical significance was set at $*p < 0.05$, $**p < 0.01$ and $***p < 0.001$. Detailed statistical information is provided in Supplementary Table 1–15.

### Reporting summary
Further information on research design is available in the Nature Portfolio Reporting Summary linked to this article.

## Data availability
All the data generated in this study are provided in the article and the Supplementary Information. The relevant raw data are provided as a Source data file. Source data are provided with this paper.

## Code availability
Raw Ca$^{2+}$ imaging data were processed using commercial software (Thinkerbiotech, Nanjing, China). The custom MATLAB scripts developed for this data processing are publicly available on Zenodo (https://doi.org/10.5281/zenodo.18279500).

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

## Acknowledgements

We thank N. Sui and J.J. Zhang for advice and comments on the manuscript; Z.Y. Qi, Z.Y. Zhang, Y. Zhang, S.Y. Liu, B.Z. Gong, and S.M. Gao for comments and advice on behavioral, functional imaging, circuit, and electrophysiological experiments; W.J. Zhou and G.C. Zou for discussions. This work was supported by the National Natural Science Foundation of China (no.82288101 to L.L., no.82301681 to X.X.L., no.82071498 to Y.X.X. and no.82471514 to Y.X.X.) and the STI2030-Major Projects (no. 2021ZD0200800 to L.L. and no. 2022ZD0214500 to Y.X.X.).

## Author contributions

Conceptualization: Y.X.X., L.L., J.S., and W.Z. Behavioral, functional imaging, circuit, and electrophysiological experiments: W.Z., X.X.L., T.S.L., Y.X.L., X.F.G., Y.F.Y., X.L., Z.W., and K.Y. Formal analysis: W.Z., X.X.L., and T.S.L. Funding acquisition: Y.X.X., L.L., and X.X.L. Project administration and supervision: Y.X.X., L.L., and J.S. Writing– original draft: W.Z., X.X.L, and T.S.L. Writing– review and editing: Y.X.X., L.L, J.S., J.W.G., and T.W.R.

## Competing interests

The authors declare no competing interests.
