## [Transparent Peer Review File · Nature Communications]

Social Reward Outcompetes Drug Seeking Dopaminergic Ensembles to Prevent Relapse

Corresponding Author: Professor Yan-Xue Xue

Version 0:

Reviewer comments:

Reviewer #1

(Remarks to the Author)

The authors investigated a very interesting effect of acute social experience decreasing cocaine relapse pressing in rats. The authors used many different tools to produce a very large amount of interesting data about the possible role of distinct groups of neurons (ensemble or cell types) in the VTA mediating this effect. While I think there are significant strengths in this paper, I do have some questions about the methods and data in the paper.

1-From the data shown in Figure 1, it appears that pressing during drug relapse decreases when social interaction was immediately before drug relapse, but not when social interaction was 4 days before drug relapse. I did not see any mention of this in Results or Discussion.

2-The “social reward treatment” they used is not a common way of assessing social behavior, which is fine but it requires some explanation and validation. It is not clear why they used juvenile rats as the social partners. Juvenile mice are used sometimes to get C57 mice to be more interested in social partners, but this is not an issue for rats that are already very social. Why not use peer rats that are age matched? The specific CPP procedure is also non-standard for social behavior and seems to be based again on what has been done previously in mice. They give a single 24 hour conditioning session with 2 juvenile male rats versus 24 hours of isolation in the opposite chamber (although I appreciate their counterbalancing the order of conditioning sessions). However, the results from this procedure could also be interpreted as avoiding the 24 hours of isolation rather than finding exposure to juvenile rats “rewarding”. This issue was a problem in some previous studies examining social CPP.

3-The authors need to validate the accuracy of the TH promoter in their AAV to obtain expression in only dopamine neurons. To my knowledge, the viral TH promoter is only 80-85% accurate most likely due to many of the restriction elements not included in the very short regulatory sequence used in this virus. For this reason, genes in these TH promoter viruses could be expressed in non-DA cells in the VTA as well. Since half the neurons in VTA are GABAergic, the TH promoter viruses could have been expressing in a significant number of non-DA GABAergic neurons.

4-Similar to the results for behavior (mentioned in comment 1), relapse-induced calcium increases assessed by photometry were attenuated following immediate exposure to social interaction, but 'not' when delayed 4 days after social. Please interpret and comment on this timing issue in paper.

5-Again, it seems that social interaction immediately before drug relapse attenuated drug-induced DA in NAc, but I don't think the authors examined whether this happens 4 days after social interaction. Please interpret and comment in paper.

6-I am well acquainted with the Fos promoter viruses but I have not read about the ER-Cre-ER, which I found is similar to the more common Cre-ERT2 gene but with an extra ERT2 component upstream of Cre-ERT2. I can see it being useful; however, the use of this virus, at least for its current purpose, needs to be validated and characterized better. For example, in Figure 2a-f, they administered 4-OHT 30 minutes 'prior' to testing for all of their many “TRAP” virus experiments. First of all, this is not a typical time point for TRAPping ensembles activated by a behavior; e.g. the original DeNardo paper uses 1 hour 'after' (not prior to) activity to inject 4-OHT and others have found even later time points are optimal, at least in mice. They don't necessarily need a full time course, but they should provide some evidence that there is both 4-OHT-dependent activation, and labeling that is induced by behavior, at 30 min prior to behavioral activation.

7-Again, there is little to no validation and characterization of most of their Fos promoter viruses in this study. These are extremely finicky viruses to get to work in brain. For example, the effectiveness of the viral Fos promoter technique is extremely sensitive to the ratio of Fos promoter-TRAP virus to its DIO-effector gene virus. Indeed, work from the van den Oever lab indicates that copy numbers of Fos 'TRAP' virus has to be much lower than copy numbers of the DIO-effector virus. Overall, the viral Fos promoter methods used here have not worked well for most people who used it to examine ensembles in higher activity neurons such as in cortex and hippocampus and even nucleus accumbens. In general, the highly variable viral copy numbers of Fos-CreERT2 virus and effector virus (e.g. Dio-mCherry) in neurons around the injection site, combined with the fact that there is always some degree of Fos promoter activation in most neurons of the above brain areas, tends to make this technique difficult at best for use. This is why I am having a great deal of trouble believing that everyone of their Fos promoter experiments worked as well as they did.

Nevertheless, I am trying to see how these Fos promoter 'TRAP' viruses might work surprisingly well in VTA neurons despite the very low rate of success in cortex or striatum. The only way that I can see these Fos promoter 'TRAP' viruses working so well here is maybe (1) because of relatively low Fos promoter activity in DA neurons inducing very low levels of Cre-ERT2 combined with (2) very different sets of DA neurons to the point of being distinctly different subtypes of DA neurons being activated by social interaction versus drug relapse. These factors could create an 'on versus off' activation between the the social versus drug relapse ensembles. One would not see this clear distinction between ensembles in regions such as cortex, hippocampus or striatum, which are made of multiple cell types.

8-I fully appreciate that they used all combinations of ensemble activation when looking at double-labeling experiments with the two different types of Fos promoter viruses. However, the fundamental problem with their comparison of social versus drug relapse ensembles here is that they are comparing two means of neural activation that cannot be directly compared: drug seeking test versus social reward, which here is just "body contact with a social partner". For drug relapse, rats have gone through extensive operant training for the drug reward, while social reward is just a single session of exposure to younger animals in a cage. Too many variables are different between the two types of reward for a fair comparison at the ensemble level using calcium imaging. For example, do we see separate ensembles because of the different sensory modalities, i.e. smelling and having contact with a social partner vs. lever pressing for a drug cue? A better comparison would be to train them for both types of reward and then see if ensembles are different when they are performing an operant response to access each separate cue. Then it would be a true test of whether we have separable social and drug ensembles, which I don't think we can really conclude that from the calcium imaging study done in Figure 2.

9-The authors also observed very distinct ensembles in VTA using single neuron calcium imaging. I can see this being true only if they are looking at the equivalent of social vs drug relapse in functionally (or perhaps genetically) distinct cell types within all DA neurons in VTA. In which case, I would more accurately call them cell types rather neuronal ensembles which are more flexible and pluripotent than cell types. None of this would have worked out this way in cortex or hippocampus where any given neuron can be involved in encoding many different memories.

10-The idea of the social ensemble suppressing the drug ensemble is very intriguing and I think there are some really interesting data in this study that support it. However, it's a bit of a problem that the TH promoter virus could be expressing in GABAergic neurons as well since its only 80-85% accurate. Cell type differences here, rather than ensemble (selected by cues) differences, could explain some of their observed differences.

11-Minor comment: The authors say they used head-fixed rats in the methods section for single neuron calcium imaging but say they used freely moving animals in page 9 of the Results section.

12-The description of viral mechanism and transport direction in the results and captions and in the diagram in panels 6a,g are the opposite of what these viruses are actually doing. AAV2/1-TH-Cre or AAV2/1-SERT-Cre is retrogradely transported from the injection site to afferent brain areas that innervate the injection site; not the other way around (not in the prograde direction). The description and figure mechanism is incorrect. Its most likely that the wording and associated figures are incorrect rather than their experiments. They need to state that AAV2/1-TH-flpo was 'retrogradely' transported from the DRN to the VTA to activate AAV-fDIO-hM4Di-EGFP in the VTA TH neurons. They need to change the direction of the arrow in their diagrams in their figures, and state that they are looking at TH projections 'from' VTA to DRN, and NOT projections from DRN to VTA. And then revise the associated description of results and interpretation throughout the manuscript.

13-In the Discussion section: "Thus, we speculate that social reward and drug seeking may represent distinct motivational states, each engaging specialized DA circuits". I actually think their data is likely saying this. However, it seems more to do with cell types or DA subtypes in VTA rather than actual ensembles selected by cues. The paper would have been amazing if they showed distinct transcriptomes or cell-type markers with ISH for the two DA cell types activated by social interaction versus drug relapse.

14- On page 21: "This antagonistic architecture" might explain why the sharp distinctions they observed with all the Fos promoter-driven viruses. But these are distinctions that you would more likely observe with distinct cell types, not with neuronal ensembles made of largely the same cell types selected by cues.

15-I think there is a typo in the caption for panel 3n: they mention 'food-responsive neuron' here and I did not see food seeking anywhere else.

Reviewer #2

(Remarks to the Author)

Zheng et al. identify two distinct dopaminergic ensembles in the VTA: one that processes social reward and another that governs drug-seeking behavior, with functional competition between the two shaping behavioral outcomes. They further delineate an organized DRN-to-VTA circuit that selectively underlies the protective effect of social reward against drug-seeking behaviors. The study's major conceptual advance lies in the identification of these two mutually inhibiting VTA dopaminergic ensembles, which not only elucidates the neuronal ensemble and circuit mechanisms through which social reward suppresses drug relapse, but also opens promising avenues for developing novel neuromodulation-based interventions for addiction. The authors skillfully employ cutting-edge methodologies to dissect the underlying neural mechanisms. Overall, the study presents compelling findings, and the manuscript is logically structured and clearly articulated. The following issues, however, should be addressed in a revision.

1. While social interaction with two juvenile conspecifics has been demonstrated to induce rewarding effects in the social conditioned place preference (sCPP) paradigm, the rigor of the present study would be significantly improved by explicitly verifying whether the 30-minute social interaction used here indeed elicits a reward state. A robust approach would be to investigate whether the identified "Social-reward-EnsDA" neurons support intracranial self-stimulation (ICSS) behavior.
2. In the Discussion, the authors cite literature suggesting that the behavioral consequences of social interaction in substance use disorder critically depend on the drug exposure history of the social partners. The present study clarifies the dopaminergic mechanisms by which interaction with drug-naïve conspecifics exerts a protective effect against drug seeking. However, the Discussion should be expanded to incorporate research on how social interaction with drug-experienced conspecifics can promote addiction. Given the growing body of evidence on the dual roles of social context in addiction—both protective and risk-promoting—the authors should integrate these perspectives to provide a more comprehensive and balanced discussion.
3. Regarding the network mapping of the ensembles: the connectivity of the drug-seeking-responsive ensemble is consistent with established evidence, as it receives inputs from the central amygdala—a structure well known to mediate Pavlovian associations of reward-predictive cues. Notably, it is surprising that neither ensemble exhibits robust inputs from the prefrontal cortex (PFC). The authors should identify and discuss prior research that supports or contextualizes this specific observation, and clarify whether this finding aligns with or challenges current models of reward and addiction circuitry.
4. Regarding the result presentation of fiber photometry recording and miniscope imaging, inconsistencies exist in the descriptions of DAergic neuron responses to social reward. Standardization of these descriptions is recommended to ensure clarity and comparability. For instance, the results from fiber photometry recording experiments are stated as: "Body contact during social interaction reliably evoked robust calcium transients in VTA DA neurons (Fig. 1h–j)." In contrast, the results from miniscope imaging experiments are described as: "17% (65 out of 380 neurons, n = 6 animals) responded selectively to social reward during the free interaction session (Fig. 2l–n and Fig. 2v)."
5. The manuscript exhibits inconsistencies in terminology formatting: the capitalization of the first letters in "Social-reward-EnsDA" and "Drug-seeking-EnsDA" is not uniform across the text, and "VTA DAergic neurons" is used interchangeably with "VTADA neurons."
6. In Fig. 1d, the label "No rewrad" is misspelled.

Reviewer #3

(Remarks to the Author)

Previous studies have shown that social rewards can help inhibit drug-seeking behavior, although the mechanisms behind this are still unclear. This study provides an intriguing perspective by identifying two populations of VTA DA neurons that are selectively activated by drug-seeking behavior and social reward, respectively. Activation of the Social-reward-Ens-DA neurons can suppress activity in the Drug-seeking-Ens-DA neurons. Additionally, these Social-reward-Ens-DA neurons receive upstream input from the DRN, a region widely recognized for its role in social behavior. Overall, this research offers a valuable mechanism to further elucidate the interaction between social and drug-related rewards in this field. Below are some concerns I have:

- 1) For Fig 1f, many green fluorescent-labeled neurons do not appear to co-label with TH. The viral vector based on the TH promoter used here is not commonly employed. Could the authors provide evidence to ensure its specificity?
- 2) In Fig 2d, the histological results only present a merged image, making it difficult to verify the co-labeling claims. It would be better to show separate channels for clarity. Additionally, why does TH expression appear so weak in D+D and S+S? Were the VTA locations for the representative images in these groups chosen consistently? Furthermore, the co-labeling observed via the cfos trap and fos staining for the same behavior seems to be quite low in the representative images.
- 3) What proportion of VTA dopamine neurons do these two populations represent? Do their downstream projections differ in a way that might explain their functional divergence?
- 4) For Fig 3e, the authors hypothesize that Social-reward-Ens-DA neurons inhibit Drug-seeking-Ens-DA neurons via non-canonical release of GABA and demonstrated a monosynaptic inhibitory connection between these two groups of dopamine neurons. However, this evidence alone is insufficient. Additional experiments such as IF or in situ should be performed to confirm co-expression of GABA markers (e.g., GAD or vGAT) and TH within Social-reward-Ens-DA neurons. Furthermore, the authors should ablate/inhibit all VTA GABAergic neurons to test whether the IPSCs still persist. Lastly, could the authors explain the observed latency in the light-evoked IPSCs?

5) I'm curious about your experiment being done under social isolation conditions. Would social interaction still inhibit relapse in group-housed mice?

6) For Figure 6, considering that AAV1 is both anterograde and retrograde, previous study (Lin et al., 2021) indicates VTADA also projects to DRN, albeit at a low proportion. Could this viral strategy inadvertently label DRN-projecting VTA DA neurons? Please clarify.

7) What major types of neurons project from DRN to the Social-reward-Ens-DA neurons? Are they DA or 5-TH neurons?

8) How do these two populations of neurons respond to other types of rewards, such as tasty food or addictive drug injections?

Minor Issues:

1) For Fig 2e, 2f, the statistical analysis should be presented as percentages rather than as raw numbers.

2) For the miniscope data, the n-value should not only represent the number of neurons but also indicate how many animals these neurons are from.

3) In Fig 5c, the starting cell is unclear, and the background appears messy.

Version 1:

Reviewer comments:

Reviewer #1

(Remarks to the Author)

The authors provided a very thorough set of good responses to my concerns. I am fine with the revised version now with one exception:

In response to my point 9, the authors quote a review by Yuste et al 2024 for their definition of a neuronal ensemble - "a population of neurons involved in a particular computation". This is only a partial definition that does not distinguish between neuronal ensemble versus cell type, and is thus insufficient. When talking about neuronal ensembles as an encoding unit, Yuste is referring to a set of neurons that are selected by a learned set of cues, and others have found are composed of multiple cell types acting together. Neuronal ensembles are not mere cell types that are involved differentially in different rewards (e.g. social versus drug reward) or (e.g. nicotine versus cocaine reward). For example in cortex, different ensembles encoding different information often have the same relative percentages of cell types.

What I was trying to say in my initial review is that you are most likely looking at different cell types for social versus drug reward that are genetically predetermined prior to learning - this means they are not neuronal ensembles in the more meaningful and informative sense. Ensembles are about encoding high-resolution learned information, and not about encoding low-resolution largely innate information that can be handled by mere cell types.

Equating neuronal ensembles with cell types is due to an incorrect over-generalization of the term neuronal ensembles. Unfortunately, this is an overly common error in the field. Please fix and I am fine with the rest of the paper, which was very interesting and a lot of good work.

Reviewer #2

(Remarks to the Author)

I have no further comments.

Reviewer #3

(Remarks to the Author)

The authors answered all my questions and I have no further comments.

Response letter to manuscript NCOMMS-25-79000-T

The authors express their gratitude to the Reviewers for providing valuable feedback. Detailed responses addressing each comment are provided below. Comments from the Reviewers are highlighted in **bold**, while author responses are presented in **plain blue text**. Modifications made to the manuscript are indicated by *italicized text on a yellow background*. Page and line numbers referenced in the responses correspond to those found in the revised manuscript.

Reviewer #1

Comments to the Author:

The authors investigated a very interesting effect of acute social experience decreasing cocaine relapse pressing in rats. The authors used many different tools to produce a very large amount of interesting data about the possible role of distinct groups of neurons (ensemble or cell types) in the VTA mediating this effect. While I think there are significant strengths in this paper, I do have some questions about the methods and data in the paper.

Response: We would like to express our sincere appreciation to the Reviewer for recognizing the value of this manuscript. We have carefully revised the manuscript and included our responses below.

1. From the data shown in Figure 1, it appears that pressing during drug relapse decreases when social interaction was immediately before drug relapse, but not when social interaction was 4 days before drug relapse. I did not see any mention of this in Results or Discussion.

Response: We sincerely appreciate the Reviewer's insightful observation that acute social interaction exerts a differential effect on drug relapse behavior depending on the interval between social exposure and the relapse test. Specifically, that lever pressing during drug relapse was reduced when social interaction was administered immediately before the test, but no such effect was observed when the interval was extended to 4 days. We apologize that this critical time-dependent characteristic of the protective effect of social rewards was not addressed in the Results and Discussion sections of the original manuscript. To rectify this oversight, we have supplemented additional experiments to systematically characterize the temporal dynamics of this protective effect, and found that the protective effect of acute social reward is transient, disappearing after 24 hours (Supplementary Fig. 18a,b).

The following text and image have been added to the manuscript:

Notably, we found that the protective effect of acute social reward (with drug-naïve partners) is transient, disappearing after 24 hours (Supplementary Fig. 18a,b). Consistently, fiber photometry recordings from the VTA and NAc revealed that drug seeking-induced calcium signal and DA release elevations were attenuated immediately following social interaction, but this attenuation was no longer observed 24 hours later (i.e., 4 days after the initial social interaction). This time-limited protective phenotype may parallel the contingency management in clinical addiction intervention³⁰: in the absence of non-drug reward reinforcers, drug-seeking behavior is highly prone to relapse^{31,32}. (page 20-21, lines 419-427 in the revised manuscript)

Supplementary Fig. 18 | The protective effect of acute social interaction against drug seeking is transient. a, Experimental design to investigate the duration of the suppressive effect of acute social reward on cocaine-seeking behavior. **b,** Number of nose pokes during cocaine self-administration training, extinction, and relapse tests. The protective effect of acute social reward against cocaine seeking was absent 24 h after social interaction ($n = 8-9$, three-way ANOVA with simple simple effect analysis: $F_{(13,3)} = 36.033$, $p < 0.001$, more statistics see Supplementary Table 13A-C).

2. The “social reward treatment” they used is not a common way of assessing social behavior, which is fine but it requires some explanation and validation. It is not clear why they used juvenile rats as the social partners. Juvenile mice are used sometimes to get C57 mice to be more interested in social partners, but this is not an issue for rats that are already very social. Why not use peer rats that are age matched? The specific CPP procedure is also non-standard for social behavior and seems to be based again on what has been done previously in mice. They give a single 24 hour conditioning session with 2 juvenile male rats versus 24 hours of isolation in the opposite chamber (although I appreciate their counterbalancing the order of conditioning sessions). However, the results from this procedure could also be interpreted as avoiding the 24 hours of isolation rather than finding exposure to juvenile rats “rewarding”. This issue was a problem in some previous studies examining social CPP.

Response: We thank the reviewer for this insightful comment. As noted, the use of juvenile conspecifics as the social reward paradigm was indeed adapted from established protocols in mice. In preliminary experiments, we compared the efficacy of age-matched peer rats versus juvenile conspecifics in inducing social CPP. While age-matched social interaction did elicit CPP in rats, the magnitude of preference was substantially weaker than that induced by juvenile conspecifics (**Supplementary Fig. 1a-d**). To further validate that our paradigm reflects social reward, we supplemented two complementary experiments:

- 1) Social self-administration assay: Rats successfully acquired the operant response to gain access to juvenile conspecifics within the first day of training (**Supplementary Fig. 1e,f**). Rapid acquisition of operant responding for a stimulus is a hallmark of its rewarding properties in behavioral neuroscience.
- 2) Optogenetic modulation of Socia-Ens activity during intracranial self-stimulation (ICSS): Activation of the Social-reward-Ens was found to enhance ICSS behavior (**Supplementary Fig. 1g-j**). This result directly links the neural ensemble encoding social reward to the brain’s core reward circuitry, confirming the rewarding nature of juvenile conspecific interaction.

The following text and image have been added to the manuscript:

In the present study, we investigate whether this similar social paradigm can serve as a social reward manipulation in rats. Behavioral assays revealed that rats exhibited a robust conditioned place preference for the context paired with juvenile conspecifics (Supplementary Fig. 1a-d) and successfully acquired the operant response required to access juvenile conspecifics (Supplementary Fig. 1e,f). Furthermore, activating the neurons activated by social interaction with juvenile conspecifics induces intracranial self-stimulation behavior (Supplementary Fig. 1g-j). Collectively, these findings confirm that interaction with juvenile conspecifics can serve as a valid experimental manipulation of social reward in rats. (page 5-6, lines 88-96 in the revised manuscript)

Supplementary Fig. 1 | Interacting with two juvenile partners is rewarding. **a**, Experimental timeline for social conditioned place preference (CPP) induced by interaction with two juvenile partners. On Day 1, rats were put in the CPP apparatus's center chamber for a 15-minute pre-conditioning test. Then, they had 24-hour social conditioning with two juvenile males in one side chamber, followed by 24-hour isolation in the other. On Day 4, they were placed in the center chamber again for a 15-minute post-conditioning test. **b**, Social CPP scores for baseline preference and the expression test. Rats exhibit a preference for the compartment associated with two juvenile partners following social CPP training ($n = 8$, paired two-tailed t test, $t_{(7)} = 9.554$, $p < 0.001$). **c**, Experimental timeline for social CPP induced by interaction with an age-matched partner. The experimental procedure was identical to that described above, except that the social stimulus was replaced with an age-matched conspecific. **d**, Social CPP scores for baseline preference and the expression test. Rats also exhibited a preference for the compartment paired with an age-matched partner following social CPP training ($n = 8$, paired two-tailed t test, $t_{(7)} = 3.200$, $p = 0.015$). **e**, Experimental timeline for social self-administration induced by interaction with two juvenile partners. Rats were trained to self-administer to gain access to social targets (two juvenile conspecifics) during daily 60-min sessions. An active nose poke triggered the automated gate to open for 60 s, allowing the rat access to the social targets in a smaller chamber. The social target chamber had an open side covered with holes spaced 1 cm apart, enabling the rats to freely interact, smell, and investigate each other without either animal entering the other chamber. An inactive nose poke, located opposite the active nose poke, had no consequences. **f**, Number of nose pokes during social self-administration sessions ($n = 8$, three-way ANOVA with simple simple effect analysis: $F_{(7,8)} = 3.781$, $p = 0.041$, more statistics see Supplementary Table12A). **g**, Virus injection strategy. **h**, Experimental timeline for optical intracranial self-stimulation (ICSS). ICSS training was conducted in operant chambers (AniLab, Ningbo, China) equipped with two nose pokes (active and inactive). An active nose poke elicited a 5-s tone and laser stimulation (1 s, 10 mW, 40 Hz, 5 ms pulse width); an inactive nose poke had no consequences. **i**, Active nose pokes at port A and port B during each 60-min ICSS session ($n = 8$, three-way ANOVA with simple simple effect analysis: day1-4, $F_{(3,12)} = 5.764$, $p = 0.011$, more statistics see Supplementary Table12B; day5-8, $F_{(3,12)} = 21.248$, $p < 0.001$, more statistics see Supplementary Table12C). **j**, Mean number of nose pokes during laser and non-laser trials, averaged across all training days. Elevated mean laser-evoked nose pokes relative to non-laser trials were observed exclusively in Chr2-expressing rats ($n = 8$, two-way ANOVA with simple effect analysis: $F_{(1,14)} = 103.654$, $p < 0.001$, more statistics see Supplementary Table12D). All the data are presented as mean \pm SEM.

3. The authors need to validate the accuracy of the TH promoter in their AAV to obtain expression in only dopamine neurons. To my knowledge, the viral TH promoter is only 80-85% accurate most likely due to many of the restriction elements not included in the very short regulatory sequence used in this virus. For this reason, genes in these TH promoter viruses could be expressed in non-DA cells in the VTA as well. Since half the neurons in VTA are GABAergic, the TH promoter viruses could have been expressing in a significant number of non-DA GABAergic neurons.

Response: We thank the reviewer for this valuable comment. We acknowledge that the TH promoter may not restrict expression exclusively to dopaminergic neurons. This is indeed a critical issue, as it bears on whether the inhibitory interaction mechanism between different ensembles is mediated by non-specific labeling of GABAergic neurons by the TH promoter, or by GABA co-release from TH-positive neurons.

To address this, we supplemented the following experiments:

- 1) We performed immunohistochemical co-labeling experiments to examine the colocalization of TRAPed neurons with Glutamate decarboxylase (GAD, a classic marker for GABAergic neurons). The results revealed that only ~10% of TRAPed neurons colocalized with GAD (**Supplementary Fig. 11a,b**).
- 2) We examined the effects of GAD antagonists on IPSCs induced by optogenetic activation of Social-reward-Ens^{DA}. The results revealed that the GAD inhibitor 3-mercaptopropionic acid (3-MPA) had no effect on IPSCs (**Supplementary Fig. 11h**).

These lines of evidence indicate that the protective effect of social reward against drug seeking is not mediated by non-specific labeling of GABAergic neurons by the TH promoter.

To clarify the mechanism underlying the protective effect of social reward against drug seeking, we reviewed relevant literature and found that Aldehyde Dehydrogenase 1a1 (ALDH1a1), but not glutamate decarboxylase (GAD), mediates a GABA synthesis pathway in midbrain DAergic neurons (Kim et al., 2015, Science; Tritsch et al., 2012, Nature). Thus, we sought to determine whether ALDH1a1 drives functional GABA co-release by Social-reward-Ens^{DA}. To this end, we supplemented the following experiments:

- 1) We performed immunohistochemical co-labeling experiments to examine the colocalization of TRAPed neurons with Aldehyde dehydrogenase 1a1 (ALDH1a1, a marker that colocalizes with TH and mediates a GABA synthesis pathway in midbrain DAergic neurons). The results revealed that ~37% of TRAPed neurons colocalized with ALDH1a1 (**Supplementary Fig. 11c,d**).
- 2) We examined the effects of the ALDH1a1 inhibitor on IPSCs induced by optogenetic activation of Social-reward-Ens^{DA}. The results revealed that the ALDH1a1 inhibitor 4-(diethylamino)-benzaldehyde (DEAB) significantly attenuated the amplitude of light-evoked IPSCs (**Supplementary Fig. 11i**).

These lines of evidence indicate that GABA co-release from TH-positive neurons, more precisely from those expressing ALDH1a1, mediates the protective effect of social reward against drug relapse.

The following text and image have been added to the manuscript:

Given that previous studies demonstrated that Aldehyde Dehydrogenase 1a1 (ALDH1a1) but not the glutamate decarboxylase (GAD) mediates a GABA synthesis pathway in midbrain DAergic neurons^{22,23}, we sought to determine whether ALDH1a1 drives functional GABA co-release by Social-reward-Ens^{DA}. To address this, we examined GAD and ALDH1a1 expression in Social-reward-Ens^{DA} via immunohistochemical co-labeling with TH. Our results revealed that only ~10% of Social-reward-Ens^{DA} expressed GAD, whereas ~37% of these neurons expressed ALDH1a1 (Supplementary Fig. 11a-d). Moreover, treatment with the ALDH inhibitor 4-(diethylamino)-benzaldehyde (DEAB, 10 μ M) but not the GAD inhibitor 3-mercaptopropionic acid (3-MPA, 500 μ M) dramatically reduced light induced IPSC amplitude (Supplementary Fig. 11e-i). Collectively, these findings establish that ALDH1a1 is required for functional GABA co-release by Social-reward-Ens^{DA}. (page 13-14, lines 265-276 in the revised manuscript)

Supplementary Fig.11 | Aldehyde Dehydrogenase 1a1 Mediates the inhibitory effect of Social-reward-Ens^{DA} on Drug-seeking-Ens^{DA}. **a**, Representative histology images of Social-reward-Ens^{DA} co-label TH or GAD. **b**, The proportion of co-labeled neurons (TH + Social-reward-Ens^{DA} vs GAD + Social-reward-Ens^{DA}) represents the total Social-reward-Ens^{DA}. **c**, Representative histology images of Social-reward-Ens^{DA} co-label TH or ALDH1a1. **d**, The proportion of co-labeled neurons (TH + Social-reward-Ens^{DA} vs ALDH1a1 + Social-reward-Ens^{DA}) represents the total Social-reward-Ens^{DA}. **e,f**, Strategy for Virus injection. **g**, Schematic of the experimental procedure for electrophysiology recording. **h**, Left: Representative traces of evoked IPSCs upon bath application of GAD inhibitor 3-mercaptopropionic acid (3-MPA, 500 μ M). Right: Amplitudes under control conditions (ACSF) or 3-MPA, normalized to baseline. **i**, Left: Representative traces of evoked IPSCs upon bath application of ALDH inhibitor 4-(diethylamino)-benzaldehyde (DEAB, 10 μ M). Right: Amplitudes under control conditions (ACSF) or DEAB, normalized to baseline (DEAB: n = 8 neurons, paired t test, **p < 0.01).

4. Similar to the results for behavior (mentioned in comment 1), relapse-induced calcium increases assessed by photometry were attenuated following immediate exposure to social interaction, but 'not' when delayed 4 days after social. Please interpret and comment on this timing issue in paper.

Response: We sincerely appreciate the Reviewer's astute observation that the photometry-based calcium response data aligns with our behavioral findings: relapse-induced calcium elevations were significantly attenuated when social interaction was administered immediately prior to the relapse test, whereas this inhibitory effect was absent when the interval between social exposure and testing was extended to 4 days (Fig. 1e-r). This cross-modal consistency (behavioral and neural recordings) corroborates the conclusion that the protective effect of social reward against drug relapse is time-dependent and transient. In response to the Reviewer's suggestion, we have incorporated a dedicated interpretation of this temporal specificity into the revised Discussion section, integrating both behavioral and neural data to highlight this timing effect.

The following text has been added to the manuscript:

Notably, we found that the protective effect of acute social reward (with drug-naïve partners) is transient, disappearing after 24 hours (Supplementary Fig. 18a,b). Consistently, fiber photometry recordings from the VTA and NAc revealed that drug seeking-induced calcium signal and DA release elevations were attenuated immediately following social interaction, but this attenuation was no longer observed 24 hours later (i.e., 4 days after the initial social interaction). This time-limited protective phenotype may parallel the contingency management in clinical addiction intervention³⁰: in the absence of non-drug reward reinforcers, drug-seeking behavior is highly prone to relapse^{31,32}. (page 20-21, lines 419-427 in the revised manuscript)

5. Again, it seems that social interaction immediately before drug relapse attenuated drug-induced DA in NAc, but I don't think the authors examined whether this happens 4 days after social interaction. Please interpret and comment in paper.

Response: We sincerely appreciate the Reviewer's critical comment regarding the need to clarify whether the modulation of NAc activity persists at the 4-day interval following social interaction. As the Reviewer astutely points out, our existing data have established a correspondence between DA release in the NAc, calcium activities in the VTA DAergic neurons, and the behavioral effects of social interaction (Fig. 1e-r and Supplementary Fig. 3d-n). To address the Reviewer's concern, we have supplemented a targeted interpretation in the revised Discussion section, explicitly stating that we did not observe any attenuation of relapse-induced DA release elevations when the interval between social interaction and the relapse test was extended to 4 days. This finding further reinforces that the regulatory effect of social reward is time-limited and transient.

The following text has been added to the manuscript:

Notably, we found that the protective effect of acute social reward (with drug-naïve partners) is transient, disappearing after 24 hours (Supplementary Fig. 18a,b). Consistently, fiber photometry recordings from the VTA and NAc revealed that drug seeking-induced calcium signal and DA release elevations were attenuated immediately following social interaction, but this attenuation was no longer observed 24 hours later (i.e., 4 days after the initial social interaction). This time-limited protective phenotype may parallel the contingency management in clinical addiction intervention³⁰: in the absence of non-drug reward reinforcers, drug-seeking behavior is highly prone to relapse^{31,32}. (page 20-21, lines 419-427 in the revised manuscript)

6. I am well acquainted with the Fos promoter viruses but I have not read about the ER-Cre-ER, which I found is similar to the more common Cre-ERT2 gene but with an extra ERT2 component upstream of Cre-ERT2. I can see it being useful; however, the use of this virus, at least for its current purpose, needs to be validated and characterized better. For example, in Figure 2a-f, they administered 4-OHT 30 minutes ‘prior’ to testing for all of their many “TRAP” virus experiments. First of all, this is not a typical time point for TRAPPING ensembles activated by a behavior; e.g. the original DeNardo paper uses 1 hour ‘after’ (not prior to) activity to inject 4-OHT and others have found even later time points are optimal, at least in mice. They don’t necessarily need a full time course, but they should provide some evidence that there is both 4-OHT-dependent activation, and labeling that is induced by behavior, at 30 min prior to behavioral activation.

Response: We sincerely appreciate the Reviewer’s insightful comments on the validation of the ER-Cre-ER viral system and the 4-OHT administration timing for TRAP experiments. We note that the canonical TRAP protocol in mice involves 4-OHT injection at 1 hour after behavioral activation, as documented in the original work by DeNardo et al. (2019, Nat Neurosci). Meanwhile, we have also taken note of the work by Wang et al. (2022, Nat Commun), which demonstrated that 4-OHT delivery 30 min prior to behavioral exposure is effective for labeling behaviorally activated neuronal ensembles in rodent models. To further validate the suitability of this protocol for our rat model, we have conducted dedicated in-house characterization experiments. Specifically, we systematically tested multiple 4-OHT injection time points relative to behavioral stimulation, and these data consistently showed that 4-OHT administration 30 min before behavioral activation yielded the highest efficiency of TRAPed neuronal labeling in our experimental setting (Supplementary Fig. 4a–d).

The following text and image have been added to the manuscript:

To validate the TRAPPING paradigm in rats, we first characterized the temporal dynamics of neuronal labeling. Rats were subjected to 30 min of social interaction at varying time intervals relative to the administration of 4-hydroxytamoxifen (4-OHT). Analysis of mCherry expression patterns demonstrated that the majority of TRAPed neurons were labeled within a 6-hour time window centered on 4-OHT injection. Notably, mCherry expression levels peaked when 4-OHT was administered 30 min prior to the initiation of social interaction (Supplementary Fig. 4a–d). (page 8, lines 151-158 in the revised manuscript)

Supplementary Fig.4 | Time course of targeted recombination in active populations (TRAPPING). a, Strategy for virus injection. b, Timeline of social stimulation experiment to determine effective TRAPPING window. c, Representative images of TRAPed cells in the ventral tegmental area (VTA) of rats subjected to the social stimulation experiment. Scale bars, 200 μ m. d, Quantification of TRAPed cell density in the VTA ($n = 4$, one-way ANOVA with bonferroni multiple comparisons test: $F_{(5,18)} = 132.4$, $p < 0.001$; home vs 3h, $p < 0.001$; home vs 0.5h, $p < 0.001$; home vs 0h, $p < 0.001$; home vs -3h, $p < 0.001$; home vs -6h, $p = 0.973$). All the data are presented as mean \pm SEM.

7. Again, there is little to no validation and characterization of most of their Fos promoter viruses in this study. These are extremely finicky viruses to get to work in brain. For example, the effectiveness of the viral Fos promoter technique is extremely sensitive to the ratio of Fos promoter-TRAP virus to its DIO-effector gene virus. Indeed, work from the van den Oever lab indicates that copy numbers of Fos ‘TRAP’ virus has to be much lower than copy numbers of the DIO-effector virus. Overall, the viral Fos promoter methods used here have not worked well for most people who used it to examine ensembles in higher activity neurons such as in cortex and hippocampus and even nucleus accumbens. In general, the highly variable viral copy numbers of Fos-CreERT2 virus and effector virus (e.g. Dio-mCherry) in neurons around the injection site, combined with the fact that there is always some degree of Fos promoter activation in most neurons of the above brain areas, tends to make this technique difficult at best for use. This is why I am having a great deal of trouble believing that everyone of their Fos promoter experiments worked as well as they did.

Nevertheless, I am trying to see how these Fos promoter ‘TRAP’ viruses might work surprisingly well in VTA neurons despite the very low rate of success in cortex or striatum. The only way that I can see these Fos promoter ‘TRAP’ viruses working so well here is maybe (1) because of relatively low Fos promoter activity in DA neurons inducing very low levels of Cre-ERT2 combined with (2) very different sets of DA neurons to the point of being distinctly different subtypes of DA neurons being activated by social interaction versus drug relapse. These factors could create an 'on versus off' activation between the the social versus drug relapse ensembles. One would not see this clear distinction between ensembles in regions such as cortex, hippocampus or striatum, which are made of multiple cell types.

Response: We sincerely appreciate the Reviewer’s critical and insightful comments regarding the validation and characterization of Fos promoter-driven TRAP viral tools, as well as the thoughtful analysis of potential factors enabling robust labeling in VTA dopaminergic neurons. We fully acknowledge that Fos promoter-based viral systems are technically demanding, with their efficacy being highly sensitive to experimental parameters, including the viral titer ratio of Fos-TRAP virus to DIO-effector virus, as highlighted by work from the van den Oever lab. We also agree that this technique often yields inconsistent results in brain regions with high basal neuronal activity (e.g., cortex, hippocampus, and striatum), due to issues such as variable viral copy number integration and non-specific Fos promoter activation in these heterogeneous cell populations.

In the present study, we optimized multiple experimental variables to ensure the specificity and efficiency of TRAP labeling in VTA dopaminergic neurons, which we believe accounts for the robust outcomes observed:

- 1) Optimization of viral titer ratio. The viral ratio used in our experiments was established based on a previous work (Lee et al., 2023, Nat Neurosci). Specifically, AAV-cFos-ERCreER (AAV2/9, 5×10^{11} vg/mL) and AAV-TH-DIO-mCherry (AAV2/8, 2×10^{13} vg/mL) were mixed at a 1:1 volume ratio for VTA injection. This ratio is consistent with the principle emphasized by the van den Oever lab that the copy number of the Fos-TRAP virus should be lower than that of the DIO-effector virus.
- 2) Temporal optimization of 4-OHT administration. As detailed in our response to Comment 6, we systematically tested multiple 4-OHT injection time points relative to behavioral stimulation. We identified that administration 30 min prior to behavior initiation yielded the peak efficiency of TRAPed neuronal labeling in the VTA (**Supplementary Fig. 4a–d**).
- 3) Stringent validation of behavior-dependent labeling specificity. To rule out non-specific Fos promoter activation, we performed a home cage control experiment where 4-OHT was administered without any behavioral manipulation. This resulted in minimal TRAP labeling in VTA neurons, confirming that the observed labeling was strictly dependent on behavioral stimulation (**Supplementary Fig. 4a–d**).
- 4) Validation of DAergic neuron-specific labeling. Colocalization analysis of TRAP-labeled neurons (mCherry⁺) with TH (a specific marker for DAergic neurons) demonstrated high co-expression efficiency, with ~90% of TRAPed neurons expressing TH (**Supplementary Fig. 11a–d**). This confirms the specificity of our labeling strategy for VTA DAergic neurons.
- 5) Distinct ensemble separation in VTA DAergic neurons. Consistent with the Reviewer’s astute hypothesis, we observed that social reward- and drug relapse-induced TRAPed neuronal populations were largely distinct, this clear separation of functional ensembles is likely attributable to two key characteristics of VTA DAergic neurons: i) relatively low basal Fos promoter activity, which minimizes non-specific Cre-ERT2 expression; and ii) the functional heterogeneity of DAergic neuron subtypes, where distinct

subpopulations are preferentially activated by social reward versus drug-associated cues. This stands in contrast to heterogeneous regions like the cortex or striatum, where overlapping activation of diverse cell types obscures ensemble specificity.

In summary, we do not intend to suggest that Fos-TRAP viral approaches are broadly applicable across brain regions, but rather that, under carefully optimized conditions, they can be selectively effective in VTA dopaminergic neurons.

The following image has been added to the manuscript:

Supplementary Fig.11 | Aldehyde Dehydrogenase 1a1 Mediates the inhibitory effect of Social-reward-Ens^{DA} on Drug-seeking-Ens^{DA}. **a**, Representative histology images of Social-reward-Ens^{DA} co-label TH or GAD. **b**, The proportion of co-labeled neurons (TH + Social-reward-Ens^{DA} vs GAD + Social-reward-Ens^{DA}) represents the total Social-reward-Ens^{DA}. **c**, Representative histology images of Social-reward-Ens^{DA} co-label TH or ALDH1a1. **d**, The proportion of co-labeled neurons (TH + Social-reward-Ens^{DA} vs ALDH1a1 + Social-reward-Ens^{DA}) represents the total Social-reward-Ens^{DA}.

8. I fully appreciate that they used all combinations of ensemble activation when looking at double-labeling experiments with the two different types of Fos promoter viruses. However, the fundamental problem with their comparison of social versus drug relapse ensembles here is that they are comparing two means of neural activation that cannot be directly compared: drug seeking test versus social reward, which here is just “body contact with a social partner”. For drug relapse, rats have gone through extensive operant training for the drug reward, while social reward is just a single session of exposure to younger animals in a cage. Too many variables are different between the two types of reward for a fair comparison at the ensemble level using calcium imaging. For example, do we see separate ensembles because of the different sensory modalities, i.e. smelling and having contact with a social partner vs. lever pressing for a drug cue? A better comparison would be to train them for both types of reward and then see if ensembles are different when they are performing an operant response to access each separate cue. Then it would be a true test of whether we have separable social and drug ensembles, which I don’t think we can really conclude that from the calcium imaging study done in Figure 2.

Response: We sincerely appreciate the Reviewer’s incisive critique regarding confounding variables in our initial ensemble comparison. We acknowledge that the divergent paradigms, that is a single social interaction session versus drug seeking behavior following extensive operant training, could attribute ensemble segregation to sensory or behavioral differences. According to your suggestions, we performed complementary operant conditioning experiments where rats were trained to nose poke for both social and cocaine rewards under matched conditions, eliminating task related disparities for a direct comparison.

Specifically, we first trained rats in Context A for social self-administration (7 consecutive days), a social active nose poke triggered the automated gate to open for 60 s, allowing the rat access to the social targets in a smaller chamber. One week later, the same cohort of rats was trained in Context B for cocaine self-administration (7 consecutive days), a drug active nose-poke triggered a ~2 s cocaine infusion. Following this matched training paradigm, we performed single neuron calcium imaging during both social seeking and drug seeking tests, where active nose pokes did not result in the delivery of any reward reinforcer (**Supplementary Fig. 9 a,b**). Our results revealed that only 10% of recorded neurons exhibited dual responsiveness to both social and cocaine cues (**Supplementary Fig. 9c–p**). Subsequent statistical analyses confirmed a lack of significant overlap between social seeking and drug seeking responsive neuronal populations (**Supplementary Fig. 9q**). These data rule out the possibility that ensemble segregation of social-seeking and drug-seeking encoding is an artifact of divergent task designs.

The following text and image have been added to the manuscript:

In addition, given the substantial array of confounding variables inherent to the social interaction versus drug relapse sessions, one might hypothesize that the observed segregation of neuronal ensembles stems from divergent sensory and behavioral modalities. To address this alternative explanation, we trained rats to perform operant responses for both social self-administration and cocaine self-administration, then assessed whether distinct neuronal ensembles were engaged when rats executed goal-directed behaviors to obtain these two discrete reward-related stimuli (Supplementary Fig. 9a,b). Our results demonstrated that only 10% of recorded neurons (26 out of 261 neurons; n = 2 animals) exhibited dual responsiveness to both social and cocaine cues (Supplementary Fig. 9c–p). Subsequent statistical analyses confirmed a lack of significant overlap between social-seeking- and drug-seeking-responsive neuronal populations (Supplementary Fig. 9q). Collectively, these findings establish that VTA DAergic ensembles exhibit stable, stimulus-specific identities and that their segregation is not a consequence of temporal factors. (page 11-12, lines 221-234 in the revised manuscript)

Supplementary Fig. 9 | Distinct VTA DAergic ensembles respond to social seeking and drug seeking. **a**, Schematic of the experimental design for microscope calcium imaging. **b**, Representative raw microscope image. **c**, **f**, **i**, **l**, Example field of view with neurons colored by the stimulus type that they are responsive to ($n = 131$ neurons). **d**, **e**, Example of a neuron selectivity responsive to cocaine seeking. **g**, **h**, Example of a neuron selectivity responsive to social seeking. **j**, **k**, Example of a neuron responsive to both. **m**, **n**, Example of a neuron responsive to neither. **o**, Matching neurons imaged across distinct sessions ($n = 261/519$ neurons). **p**, Percentage of neurons responsive to social seeking, cocaine seeking, both, or neither ($n = 261$ neurons). **q**, Overlap compared with null distribution assuming Social-seeking- and Drug-seeking-responsive neurons are independent samples. All the data are presented as mean \pm SEM.

9. The authors also observed very distinct ensembles in VTA using single neuron calcium imaging. I can see this being true only if they are looking at the equivalent of social vs drug relapse in functionally (or perhaps genetically) distinct cell types within all DA neurons in VTA. In which case, I would more accurately call them cell types rather neuronal ensembles which are more flexible and pluripotent than cell types. None of this would have worked out this way in cortex or hippocampus where any given neuron can be involved in encoding many different memories.

Response: We sincerely appreciate the Reviewer's insightful comment regarding the distinction between neuronal ensembles and cell types in the VTA. We acknowledge that the two dopaminergic populations responsive to social reward and drug seeking may exhibit distinct genetic profiles, and further validation of cell type specificity via techniques such as single cell sequencing would represent an important and interesting direction for future research.

In here, our use of the term "neuronal ensemble" in the present study is grounded in two lines of published evidence in the field:

- 1) The review (Yuste et al., 2024, Neuron) defined a neuronal ensemble as "a population of neurons involved in a particular computation". The notion of ensemble implies that coding is produced by populations of neurons whose individual contributions are noisy but that together produce coherent outputs. The term is mostly used within systems and computational neuroscience to describe a neural network with a particular function.
- 2) The study (Gimenez-Gomez et al., 2025, Nat Neurosci) validated the existence of binge drinking associated neuronal ensembles in the mouse medial orbitofrontal cortex (mOFC) by combining TRAP labeling with cFos immunofluorescence staining. They demonstrated that neurons TRAPed during the first binge drinking episode could be reactivated during a subsequent binge drinking challenge, with a colocalization rate of 50%–60%. This stable reactivation confirmed that binge alcohol drinking recruits a dedicated neuronal ensemble in the mOFC.

To confirm that the TRAP labeled VTA dopaminergic neurons constitute social reward ensembles and drug seeking ensembles, we performed analogous experiments. Specifically, we combined TRAP labeling with cFos immunofluorescence staining to examine whether neurons labeled during an initial episode of drug seeking or social interaction could be reactivated during a second temporally separated exposure to the same stimulus. Our results showed that 50%–60% of the labeled neurons were reactivated across the two distinct time points of the same stimulus exposure (**Supplementary Fig. 5j-n**). Thus, by using the term "ensemble," we intend to describe the coordinated, synchronous activity of these neurons that collectively drives a specific behavioral output (social reward vs. drug seeking).

Finally, we have revised our Results and Discussion to address this important distinction, we view "cell type" as a static identity and "ensemble" as the active functional state of these neurons during behavior. We hope these clarifications and the revised manuscript adequately address the Reviewer's conceptual concern.

The following text has been added to the manuscript:

Neuronal ensembles, defined as groups of neurons exhibiting recurring patterns of coordinated activity, represent a functional unit intermediate between single neurons and broader brain regions²¹. To determine whether the TRAPed neurons in the VTA constitute a neuronal ensemble, we examined whether neurons labeled during the first episode of drug seeking or social interaction could be reactivated during a second, temporally separated episode of the same stimulus. Indeed, re-exposure to the same condition robustly reactivated TRAPed neurons, as evidenced by their overlap with cFos expression, confirming the presence of stable Drug-seeking- and Social-reward-responsive ensembles within the VTA (Supplementary Fig. 5l-s). (page 9-10, lines 173-181 in the revised manuscript)

However, it remains unclear whether these two ensembles exhibit distinct transcriptomic profiles or cell-type-specific markers, and future studies employing techniques such as single-cell sequencing and spatial transcriptomics to characterize their transcriptional features and molecular markers represent a promising and interesting direction for further investigation. (page 22, lines 457-462 in the revised manuscript)

10. The idea of the social ensemble suppressing the drug ensemble is very intriguing and I think there are some really interesting data in this study that support it. However, it's a bit of a problem that the TH promoter virus could be expressing in GABAergic neurons as well since its only 80-85% accurate. Cell type differences here, rather than ensemble (selected by cues) differences, could explain some of their observed differences.

Response: We sincerely appreciate the Reviewer's insightful comment regarding the potential off-target expression of the TH promoter-driven virus and its implications for interpreting our ensemble-specific findings. We fully acknowledge that it is necessary to rule out the possibility that TH promoter-based viruses may exhibit non-specific expression in GABAergic neurons. To address this, we performed two complementary lines of investigation to dissociate the roles of DA-mediated GABA co-release versus traditional GABAergic transmission:

- 1) We performed immunohistochemical co-labeling experiments to examine the colocalization of TRAPed neurons with Glutamate decarboxylase (GAD, a classic marker for GABAergic neurons). The results revealed that only ~10% of TRAPed neurons colocalized with GAD (**Supplementary Fig. 11a,b**), suggesting while a small degree of off-target expression is inherent to the TH promoter, this low percentage indicates that the majority of our manipulated cells are indeed dopaminergic neurons.
- 2) We examined the effects of GAD antagonists on IPSCs induced by optogenetic activation of Social-reward-Ens^{DA}. The results revealed that the GAD inhibitor 3-mercaptopropionic acid (3-MPA) had no significant effect on IPSCs (**Supplementary Fig. 11h**).

These lines of evidence indicate that the protective effect of social reward against drug seeking is not mediated by non-specific labeling of GABAergic neurons by the TH promoter.

Given that previous studies (Kim et al., 2015, Science; Tritsch et al., 2012, Nature) revealed that Aldehyde Dehydrogenase 1a1 (ALDH1a1), but not glutamate decarboxylase (GAD), mediates a GABA synthesis pathway in midbrain DAergic neurons. We sought to determine whether ALDH1a1 drives functional GABA co-release by Social-reward-Ens^{DA}. To this end, we supplemented the following experiments:

- 1) We performed immunohistochemical co-labeling experiments to examine the colocalization of TRAPed neurons with Aldehyde dehydrogenase 1a1 (ALDH1a1, a marker that colocalizes with TH and mediates a GABA synthesis pathway in midbrain DAergic neurons). The results revealed that ~37% of TRAPed neurons colocalized with ALDH1a1 (**Supplementary Fig. 11c,d**).
- 2) We examined the effects of the ALDH1a1 inhibitor on IPSCs induced by optogenetic activation of Social-reward-Ens^{DA}. The results revealed that the ALDH1a1 inhibitor 4-(diethylamino)-benzaldehyde (DEAB) significantly attenuated the amplitude of light-evoked IPSCs (**Supplementary Fig. 11i**).

These lines of evidence indicate that GABA co-release from TH-positive neurons, more precisely from those expressing ALDH1a1, mediates the protective effect of social reward against drug relapse.

In summary, by demonstrating that the inhibitory effect remains intact even when GAD-mediated GABA synthesis is blocked, we have reinforced the conclusion that the suppression of the drug-seeking ensemble is a specific functional property of the recruited DAergic ensembles.

The following text and image have been added to the manuscript:

Given that previous studies demonstrated that Aldehyde Dehydrogenase 1a1 (ALDH1a1) but not the glutamate decarboxylase (GAD) mediates a GABA synthesis pathway in midbrain DAergic neurons^{22,23}, we sought to determine whether ALDH1a1 drives functional GABA co-release by Social-reward-Ens^{DA}. To address this, we examined GAD and ALDH1a1 expression in Social-reward-Ens^{DA} via immunohistochemical co-labeling with TH. Our results revealed that only ~10% of Social-reward-Ens^{DA} expressed GAD, whereas ~37% of these neurons expressed ALDH1a1 (Supplementary Fig. 11a-d). Moreover, treatment with the ALDH inhibitor 4-(diethylamino)-benzaldehyde (DEAB, 10 μ M) but not the GAD inhibitor 3-mercaptopropionic acid (3-MPA, 500 μ M) dramatically reduced light induced IPSC amplitude (Supplementary Fig. 11e-i). Collectively, these findings establish that ALDH1a1 is required for functional GABA co-release by Social-reward-Ens^{DA}. (page 13-14, lines 265-276 in the revised manuscript)

11. Minor comment: The authors say they used head-fixed rats in the methods section for single neuron calcium imaging but say they used freely moving animals in page 9 of the Results section.

Response: We sincerely appreciate the Reviewer for pointing out this inconsistency in our manuscript. To rectify this error, we have revised the relevant content in the Methods section to accurately reflect the freely moving imaging paradigm employed in the study.

The following text has been added to the manuscript:

After ~3 weeks of recovery, a baseplate was aligned and fixed with dental cement. Subsequent single-photon imaging sessions were conducted during behavioral tasks, including social interaction and drug relapse tests. (page 49, lines 853-855 in the revised manuscript)

12. The description of viral mechanism and transport direction in the results and captions and in the diagram in panels 6a,g are the opposite of what these viruses are actually doing. AAV2/1-TH-Cre or AAV2/1-SERT-Cre is retrogradely transported from the injection site to afferent brain areas that innervate the injection site; not the other way around (not in the prograde direction). The description and figure mechanism is incorrect. Its most likely that the wording and associated figures are incorrect rather than their experiments. They need to state that AAV2/1-TH-flpo was 'retrogradely' transported from the DRN to the VTA to activate AAV-fDIO-hM4Di-EGFP in the VTA TH neurons. They need to change the direction of the arrow in their diagrams in their figures, and state that they are looking at TH projections 'from' VTA to DRN, and NOT projections from DRN to VTA. And then revise the associated description of results and interpretation throughout the manuscript.

Response: We sincerely appreciate the Reviewer's critical comment. We apologize for the confusion caused by the imprecise wording in the original version.

- 1) First, we clarify that the AAV-SERT-Cre we employed is based on the AAV2/9 serotype, which does not possess trans-synaptic transport capabilities, and this has been revised in the figures.
- 2) Second, our viral approach relies on the anterograde trans-synaptic properties of AAV2/1, which enables monosynaptic tracing and modulation according to previous studies (Zingg B et.al, 2017, Neuron; Qi G et.al, 2022, Nat Commun). The detailed mechanism is as follows: We first injected AAV2/1-TH-Flpo into the DRN. Adeno-associated virus AAV2/1 has been shown to undergo anterograde monosynaptic transmission; thus, AAV2/1-TH-Flpo is anterograde transported monosynaptically to postsynaptic dopaminergic neurons in the VTA, where Flpo is expressed. In the VTA, we co-injected AAV-FDIO-hM3Dq-EGFP (a Flpo-dependent vector), ensuring that hM3Dq-EGFP is only expressed in VTA dopaminergic neurons receiving monosynaptic inputs from DRN neurons labeled by AAV2/1-TH-Flpo.
- 3) Finally, to rule out the effect of potential retrograde transport of AAV2/1, we added a control group where we artificially activated VTA-DRN dopaminergic projections and found no significant effect on drug seeking (Supplementary Fig. 17f-j).

The following text and image have been added to the manuscript:

To test this, we combined an anterograde monosynaptic virus (AAV2/1-TH-flpo) with a flpo-dependent DREADD virus (AAV-FDIO-hM3Dq-EGFP) to selectively activate VTA DAergic neurons that receive input from the DRN (Fig. 6a,b and Supplementary Fig. 15a,b). (page 18, lines 371-374 in the revised manuscript)

To rule out the effect of potential retrograde transport of AAV2/1-Cre²⁷ and make our conclusion more convincing, we established control groups to dissect the functional roles of DRN-VTA projections versus VTA-DRN projections. The results showed that activation of the DRN-VTA serotonergic projections, rather than VTA-DRN dopaminergic projections, mimicked the therapeutic effect of social reward against drug seeking (Supplementary Fig. 17a-j), confirming the critical role of DRN serotonergic inputs. (page 19, lines 383-390 in the revised manuscript)

Supplementary Fig.17 | Activation of the serotonergic DRN-VTA but not VTA-DRN projections mimicked the therapeutic effect of social reward against cocaine seeking. f, Left: Strategy for Virus injection. Right:

Representative photo of hM3Dq-mCherry expression in the DRN. Scale bar: 500 μ m. **g**, Experimental scheme to investigate the effect of the VTA-DRN dopaminergic projection on cocaine-seeking behavior. **h**, Number of nose-pokes during the 10-day cocaine self-administration training. **i**, Number of nose-pokes during the 14-day extinction training. **j**, Number of nose pokes during the relapse test. Activation of the VTA-DRN dopaminergic projection had no effect on cocaine-seeking behavior during the relapse test ($n = 8$, two-way ANOVA with simple effect analysis: $F_{(1,14)} = 1.163$, $p = 0.299$, more statistics see Supplementary Table 8J-L). All data are presented as mean \pm SEM.

13. In the Discussion section: “Thus, we speculate that social reward and drug seeking may represent distinct motivational states, each engaging specialized DA circuits”. I actually think their data is likely saying this. However, it seems more to do with cell types or DA subtypes in VTA rather than actual ensembles selected by cues. The paper would have been amazing if they showed distinct transcriptomes or cell-type markers with ISH for the two DA cell types activated by social interaction versus drug relapse.

Response: We sincerely appreciate the Reviewer’s insightful comment. We agree that the two dopaminergic populations responsive to social reward and drug seeking may exhibit distinct genetic profiles or molecular markers, and further validation of cell type specificity via techniques such as single-cell sequencing and transcriptomic analysis would represent an important and interesting direction for future research. Accordingly, we have supplemented the Discussion section with relevant statements to address this point.

The following text has been revised to the manuscript:

Here, we demonstrate that social reward and drug seeking engage minimally overlapping DAergic ensembles, and they establish different connections with brain regions. These findings are consistent with previous studies and further support the unique role of distinct DAergic population^{42,43}. However, it remains unclear whether these two ensembles exhibit distinct transcriptomic profiles or cell-type-specific markers, and future studies employing techniques such as single-cell sequencing and spatial transcriptomics to characterize their transcriptional features and molecular markers represent a promising and interesting direction for further investigation. (page 22, lines 454-462 in the revised manuscript)

14. On page 21: “This antagonistic architecture” might explain why the sharp distinctions they observed with all the Fos promoter-driven viruses. But these are distinctions that you would more likely observe with distinct cell types, not with neuronal ensembles made of largely the same cell types selected by cues.

Response: We are grateful for the Reviewer’s thoughtful critique regarding the conceptual distinction between "cell types" and "neuronal ensembles." This is a fundamental question in systems neuroscience, and the Reviewer’s point regarding the "sharp distinctions" we observed is well-taken. We agree that the line between a highly specialized cell type and a stable functional ensemble can be nuanced, especially in the VTA where molecular and circuit heterogeneity is prominent.

In light of the Reviewer’s feedback, we have carefully re-evaluated our terminology. We do not intend to suggest that these neurons are randomly recruited from a homogeneous pool. Instead, we propose that Social-reward-Ens and Drug-seeking-Ens represent functional units that are "nested" within existing molecular and circuit architectures. We have adopted the term "ensemble" primarily to emphasize the experience-dependent recruitment and functional synergy that defines their role in behavior.

We clarify our use of "neuronal ensemble" with two lines of evidence:

- 1) We ruled out contributions from non-DA cell types (especially GABAergic neurons) via complementary experiments: Immunohistochemical co-labeling showed only ~10% of TRAPed neurons colocalized with GAD (a GABAergic marker), and patch-clamp recordings demonstrated that the GAD inhibitor 3-mercaptopropionic acid (3-MPA) had no significant effect on IPSCs induced by optogenetic activation of Social-reward-Ens^{DA} (**Supplementary Fig. 11a-i**). These results confirm our observed effects are mediated by DAergic populations.
- 2) Our terminology aligns with recent field definitions. The review (Yuste et al., 2024, Neuron) defined a neuronal ensemble as “groups of neurons displaying recurring patterns of coordinated activity”, which is consistent with our framework. Similarly, the study (Gimenez-Gomez et al., 2025, Nat Neurosci) used this approach to describe how specific rewards organize cortical neurons into stable, behaviorally relevant groups.

Of course, we acknowledge that the two DAergic ensembles may exhibit distinct genetic profiles. Validating more refined cell-type classifications via techniques like single-cell sequencing and spatial transcriptomics represents an important future direction. To address the Reviewer’s concern, we have removed the phrase “This antagonistic architecture” in full and revised the Discussion to explicitly note that these functional ensembles may be anchored in underlying refined cell-type identities.

The following text has been revised to the manuscript:

However, it remains unclear whether these two ensembles exhibit distinct transcriptomic profiles or cell-type-specific markers, and future studies employing techniques such as single-cell sequencing and spatial transcriptomics to characterize their transcriptional features and molecular markers represent a promising and interesting direction for further investigation. (page 22, lines 457-462 in the revised manuscript)

15. I think there is a typo in the caption for panel 3n: they mention ‘food-responsive neuron’ here and I did not see food seeking anywhere else.

Response: We sincerely appreciate the Reviewer for pointing out this typo. We have corrected the error in the caption of panel 3n.

The following text has been revised to the manuscript:

***n**, Changes of magnitude in response to drug-seeking after-before social reward experience (n = 110 neurons)*
(page 32, lines 590-592 in the revised manuscript)

Reviewer #2

Comments to the Author:

Zheng et al. identify two distinct dopaminergic ensembles in the VTA: one that processes social reward and another that governs drug-seeking behavior, with functional competition between the two shaping behavioral outcomes. They further delineate an organized DRN-to-VTA circuit that selectively underlies the protective effect of social reward against drug-seeking behaviors. The study's major conceptual advance lies in the identification of these two mutually inhibiting VTA dopaminergic ensembles, which not only elucidates the neuronal ensemble and circuit mechanisms through which social reward suppresses drug relapse, but also opens promising avenues for developing novel neuromodulation-based interventions for addiction. The authors skillfully employ cutting-edge methodologies to dissect the underlying neural mechanisms. Overall, the study presents compelling findings, and the manuscript is logically structured and clearly articulated. The following issues, however, should be addressed in a revision.

Response: Thank you for reviewing our manuscript and providing valuable constructive suggestions, which have greatly improved its quality. The questions and comments have been carefully considered during the revision process, and we have provided detailed responses to each inquiry below.

1. While social interaction with two juvenile conspecifics has been demonstrated to induce rewarding effects in the social conditioned place preference (sCPP) paradigm, the rigor of the present study would be significantly improved by explicitly verifying whether the 30-minute social interaction used here indeed elicits a reward state. A robust approach would be to investigate whether the identified "Social-reward-EnsDA" neurons support intracranial self-stimulation (ICSS) behavior.

Response: We sincerely appreciate the Reviewer’s valuable comment regarding the need to verify the rewarding nature of the 30-minute social interaction paradigm. In response, we supplemented two complementary experiments to address this critical point:

- 1) **Social self-administration assay:** our results revealed that rats successfully acquired the operant response to gain access to juvenile conspecifics within the first day of training (**Supplementary Fig. 1e,f**). Rapid acquisition of operant responding for a stimulus is a well-recognized hallmark of its rewarding properties in behavioral neuroscience.
- 2) **Optogenetic modulation of Social-reward-Ens during intracranial self-stimulation (ICSS):** we found that optogenetic activation of the social engram induced ICSS behavior (**Supplementary Fig. 1g-j**). These findings directly link the neural ensemble encoding social reward to the brain’s core reward circuitry, thereby confirming the rewarding nature of interaction with juvenile conspecifics.

The following text and image have been added to the manuscript:

In the present study, we first investigate whether this similar social paradigm can serve as a social reward manipulation in rats. Behavioral assays revealed that rats exhibited a robust conditioned place preference for the context paired with juvenile conspecifics (Supplementary Fig. 1a-d) and successfully acquired the operant response required to access juvenile conspecifics (Supplementary Fig. 1e,f). Furthermore, activating the neurons activated by social interaction with juvenile conspecifics induces intracranial self-stimulation behavior (Supplementary Fig. 1g-j). Collectively, these findings confirm that interaction with juvenile conspecifics can serve as a valid experimental manipulation of social reward in rats. (page 5-6, lines 88-96 in the revised manuscript)

Supplementary Fig. 1 | Interacting with two juvenile partners is rewarding. *e*, Experimental timeline for social self-administration induced by interaction with two juvenile partners. Rats were trained to self-administer to gain access to social targets (two juvenile conspecifics) during daily 60-min sessions. An active nose poke triggered the automated gate to open for 60 s, allowing the rat access to the social targets in a smaller chamber. The social target chamber had an open side covered with holes spaced 1 cm apart, enabling the rats to freely interact, smell, and investigate each other without either animal entering the other chamber. An inactive nose poke, located opposite the active nose poke, had no consequences. *f*, Number of nose pokes during social self-administration

sessions ($n = 8$, three-way ANOVA with simple simple effect analysis: $F_{(7,8)} = 3.781$, $p = 0.041$, more statistics see Supplementary Table12A). **g**, Virus injection strategy. **h**, Experimental timeline for optical intracranial self-stimulation (ICSS). ICSS training was conducted in operant chambers (AniLab, Ningbo, China) equipped with two nose pokes (active and inactive). an active nose poke elicited a 5-s tone and laser stimulation (1 s, 10 mW, 40 Hz, 5 ms pulse width); an inactive nose poke had no consequences. **i**, Active nose pokes at port A and port B during each 60-min ICSS session ($n = 8$, three-way ANOVA with simple simple effect analysis: day1-4, $F_{(3,12)} = 5.764$, $p = 0.011$, more statistics see Supplementary Table12B; day5-8, $F_{(3,12)} = 21.248$, $p < 0.001$, more statistics see Supplementary Table12C). **j**, Mean number of nose pokes during laser and non-laser trials, averaged across all training days. Elevated mean laser-evoked nose pokes relative to non-laser trials were observed exclusively in ChR2-expressing rats ($n = 8$, two-way ANOVA with simple effect analysis: $F_{(1,14)} = 103.654$, $p < 0.001$, more statistics see Supplementary Table12D). All the data are presented as mean \pm SEM.

2. In the Discussion, the authors cite literature suggesting that the behavioral consequences of social interaction in substance use disorder critically depend on the drug exposure history of the social partners. The present study clarifies the dopaminergic mechanisms by which interaction with drug-naïve conspecifics exerts a protective effect against drug seeking. However, the Discussion should be expanded to incorporate research on how social interaction with drug-experienced conspecifics can promote addiction. Given the growing body of evidence on the dual roles of social context in addiction—both protective and risk-promoting—the authors should integrate these perspectives to provide a more comprehensive and balanced discussion.

Response: We sincerely appreciate the Reviewer's valuable comment. In response to your suggestion, we have expanded the Discussion section to incorporate the dual roles of social context in addiction (protective vs. risk-promoting) and integrated relevant research on social interaction with drug-experienced conspecifics, aiming to provide a more comprehensive and balanced perspective.

The following text has been added to the manuscript:

Accumulating evidence indicates that DAergic neurons may mediate the dual effects of social interaction on SUD⁵. A recent study demonstrated that corticotropin-releasing hormone release from the piriform cortex to amygdala is selectively enhanced during negative social interaction (with drug-exposed partners)³³. In contrast, our study showed that the DRN-VTA pathway is selectively enhanced during social reward (with drug-naïve partners); specifically, Drug-seeking-Ens^{DA} receives greater input from the amygdala, whereas Social-reward-Ens^{DA} receives greater input from the DRN. Thus, we hypothesize that the protective effect of social reward is mediated by activation of the DRN-VTA pathway, whereas the opposing effect of negative social interaction may arise from excessive activation of the amygdala-VTA pathway. (page 21, lines 432-441 in the revised manuscript)

3. Regarding the network mapping of the ensembles: the connectivity of the drug-seeking-responsive ensemble is consistent with established evidence, as it receives inputs from the central amygdala—a structure well known to mediate Pavlovian associations of reward-predictive cues. Notably, it is surprising that neither ensemble exhibits robust inputs from the prefrontal cortex (PFC). The authors should identify and discuss prior research that supports or contextualizes this specific observation, and clarify whether this finding aligns with or challenges current models of reward and addiction circuitry.

Response: We sincerely appreciate the Reviewer's insightful comment regarding the unexpected lack of robust prefrontal cortex (PFC) inputs to the VTA DAergic ensembles and the need to contextualize this observation with prior literature.

We share the initial surprise about the relatively sparse inputs from the infralimbic (IL) and prelimbic (PL) cortices to the DAergic ensembles identified in our study. To address this, we reviewed relevant literature and found consistent evidence supporting our observation: Watabe-Uchida et al. (2012) performed comprehensive whole-brain circuit tracing of direct inputs to midbrain DAergic neurons and reported that, compared to other brain regions (e.g., amygdala, dorsal raphe nucleus), the number and proportion of inputs from the IL and PL cortices to DAergic neurons in the VTA are indeed relatively low. Thus, our finding aligns with this established literature and does not challenge current models of reward and addiction circuitry. Furthermore, we have added relevant statements in the discussion section. Moreover, we have supplemented the relevant discussion section to clarify this point.

The following image has been added to the manuscript:

Here, we demonstrate that social reward and drug seeking engage minimally overlapping DAergic ensembles, and they establish different connections with brain regions. These findings are consistent with previous studies and further support the unique role of distinct DAergic population^{44,45}. (page 22, lines 454-457 in the revised manuscript)

4. Regarding the result presentation of fiber photometry recording and miniscope imaging, inconsistencies exist in the descriptions of DAergic neuron responses to social reward. Standardization of these descriptions is recommended to ensure clarity and comparability. For instance, the results from fiber photometry recording experiments are stated as: “Body contact during social interaction reliably evoked robust calcium transients in VTA DA neurons (Fig. 1h–j).” In contrast, the results from miniscope imaging experiments are described as: “17% (65 out of 380 neurons, n = 6 animals) responded selectively to social reward during the free interaction session (Fig. 2l–n and Fig. 2v).”

Response: We sincerely appreciate the Reviewer’s careful observation regarding the inconsistent descriptions of DAergic neuron responses to social reward across fiber photometry and miniscope imaging results. Specifically, we have updated the description of miniscope imaging results to align with the wording used for fiber photometry data, emphasizing body contact during social interaction as the common stimulus for both assays.

The following text has been added to the manuscript:

17% (65 out of 380 neurons, n = 6 animals) responded selectively to body contact during social interaction (Fig. 2l–n and Fig. 2v). (page 10, lines 200–201 in the revised manuscript)

5. The manuscript exhibits inconsistencies in terminology formatting: the capitalization of the first letters in “Social-reward-EnsDA” and “Drug-seeking-EnsDA” is not uniform across the text, and “VTA DAergic neurons” is used interchangeably with “VTADA neurons.”

Response: We sincerely appreciate the Reviewer for pointing out the inconsistencies in terminology formatting. We have carefully reviewed the entire manuscript, standardized the capitalization of "Social-reward-Ens^{DA}" and "Drug-seeking-Ens^{DA}" (ensuring uniform capitalization throughout), and unified the terminology to "VTA DAergic neurons" (replacing all instances of "VTA^{DA} neurons") for consistency.

6. In Fig. 1d, the label "No rewrad" is misspelled.

Response: We sincerely appreciate the Reviewer for pointing out this typo. We have corrected the misspelling "No rewrad" to "No reward" in Figure 1d to ensure accuracy.

Reviewer #3

Comments to the Author:

Previous studies have shown that social rewards can help inhibit drug-seeking behavior, although the mechanisms behind this are still unclear. This study provides an intriguing perspective by identifying two populations of VTA DA neurons that are selectively activated by drug-seeking behavior and social reward, respectively. Activation of the Social-reward-Ens-DA neurons can suppress activity in the Drug-seeking-Ens-DA neurons. Additionally, these Social-reward-Ens-DA neurons receive upstream input from the DRN, a region widely recognized for its role in social behavior. Overall, this research offers a valuable mechanism to further elucidate the interaction between social and drug-related rewards in this field. Below are some concerns I have:

Response: Thank you for taking the time to review our manuscript and providing valuable feedback, which has greatly enhanced the quality of this paper. We have carefully incorporated your suggestions into the revised version, and our responses are provided below.

Here are some specific comments:

1. For Fig 1f, many green fluorescent-labeled neurons do not appear to co-label with TH. The viral vector based on the TH promoter used here is not commonly employed. Could the authors provide evidence to ensure its specificity?

Response: We sincerely appreciate the Reviewer's critical comment regarding the specificity of the TH promoter driven viral vector and the apparent lack of co labeling in Figure 1f. We apologize for the ambiguity caused by not presenting individual fluorescent channels in the original figure, which obscured the true co localization efficiency. To clarify this point, we have now provided separate fluorescent channel images for Fig 1f, clearly showing the overlap between green fluorescent labeled neurons and TH positive neurons. Quantitative analysis further confirms that over 90% of green fluorescent labeled neurons co express TH, demonstrating the high specificity of our TH promoter based viral vector for targeting VTA DAergic neurons.

The following image has been added to the manuscript:

Fig.1 | The VTA DAergic neurons act as a central nexus of social reward and drug seeking. e.f. Representative photo of GCamp6s expression in the VTA. Green: GCamp6s. Pink: Tyrosine hydroxylase, TH.

2. In Fig 2d, the histological results only present a merged image, making it difficult to verify the co-labeling claims. It would be better to show separate channels for clarity. Additionally, why does TH expression appear so weak in D+D and S+S? Were the VTA locations for the representative images in these groups chosen consistently? Furthermore, the co-labeling observed via the cfos trap and fos staining for the same behavior seems to be quite low in the representative images.

Response: We sincerely appreciate the Reviewer's detailed and insightful comments regarding Figure 2d. In response to your suggestions, we have implemented the following revisions to address the concerns raised:

- 1) **Separate fluorescent channel display:** To improve the clarity of co-labeling verification, we have added individual channel images corresponding to Fig 2d in the supplementary materials (**Supplementary Fig.5d-o**) of the revised manuscript.
- 2) **Optimization of representative images and tissue section selection:** We carefully reviewed the original experimental data and reselected representative images for the D+D and S+S groups based on multiple criteria including fluorescent intensity of TH expression, co-labeling efficiency and anatomical positional consistency. The previously selected images were not sufficiently representative, leading to the misleading appearance of weak TH expression and low co-labeling. The updated representative images (**Supplementary Fig.5d-o**) more accurately reflect the true experimental results. Moreover, given that TRAP labeling and cFos activation exhibit slight positional differences during social interaction and drug relapse sessions, we selected tissue sections from adjacent regions with minimal distance where both signals are robust, which ensures anatomical validity while optimizing the visualization of co-labeling patterns.

The following image has been added to the manuscript:

Supplementary Fig. 5 | Social-reward- and Drug-seeking-responsive DAergic ensembles in the VTA are mainly distinct. **d**, Representative images of TH staining in the Social-reward-responsive ensembles (Label 1) and Drug-seeking-responsive ensembles (Label 2). Red: mCherry-expressed positive neurons (Label 1); Green: cfos-expressed positive neurons (Label 2); Pink: TH-expressed positive neurons; Blue: DAPI. Scale bar: 50 μ m. **e**, Percentage of co-labeled neurons in Social-reward-responsive ensembles (Label 1) or Drug-seeking-responsive ensembles (Label 2) ($n = 5$). **f**, Percentage of co-labeled DAergic neurons in Social-reward-responsive ensembles (Label 1) or Drug-seeking-responsive ensembles (Label 2) ($n = 5$). **g**, Representative images of tyrosine hydroxylase (TH) staining in the Drug-seeking-responsive ensembles (Label 1) and Social-reward-responsive ensembles (Label 2). **h**, Percentage of co-labeled neurons in Drug-seeking-responsive ensembles (Label 1) or Social-reward-responsive ensembles (Label 2) ($n = 4$). **i**, Percentage of co-labeled DAergic neurons in Drug-seeking-responsive ensembles (Label 1) or Social-reward-responsive ensembles (Label 2) ($n = 4$). **g**, Representative images of tyrosine hydroxylase (TH) staining in the Drug-seeking-responsive ensembles (Label 1 versus Label 2). **k**, Percentage of co-labeled neurons in Drug-seeking-responsive ensembles (Label 1 versus Label 2) ($n = 4$). **l**, Percentage of co-labeled DAergic neurons in Drug-seeking-responsive ensembles (Label 1 versus Label 2) ($n = 4$). **m**, Representative images of tyrosine hydroxylase (TH) staining in the Social-reward-responsive ensembles (Label 1 versus Label 2). **n**, Percentage of co-labeled neurons in Social-reward-responsive ensembles (Label 1 versus Label 2) ($n = 4$). **o**, Percentage of co-labeled DAergic neurons in Social-reward-responsive ensembles (Label 1 versus Label 2) ($n = 4$). All the data are presented as mean \pm SEM.

3. What proportion of VTA dopamine neurons do these two populations represent? Do their downstream projections differ in a way that might explain their functional divergence?

Response: We sincerely appreciate the Reviewer's insightful questions regarding the proportion of VTA DAergic neurons represented by the two ensembles and the potential functional implications of their downstream projections. In response, we have supplemented quantitative data on the ensemble proportions and expanded the Discussion section to address downstream projection divergence, as detailed below:

- 1) We calculated the proportion of Social-reward-Ens^{DA} and Drug-seeking-Ens^{DA} relative to total VTA DAergic neurons. The results, consistent with the results of single-cell calcium imaging (Fig.2v, Social-reward-Ens^{DA}:31%, Drug-seeking-Ens^{DA}:26%; Supplementary Fig.9p, Social-seeking-Ens^{DA}:33%, Drug-seeking-Ens^{DA}:35%), show that both ensembles account for approximately 30% of total VTA DAergic neurons (**Supplementary Fig. 6d,e**).
- 2) We agree that divergent downstream projections are likely critical for explaining the functional differences between the two ensembles. We have therefore added relevant discussion to contextualize this with existing literature and outline future research directions.

The following text and image have been added to the manuscript:

Moreover, the number of DAergic neurons activated during social reward versus drug relapse sessions did not differ significantly; both populations accounted for approximately 30% of total DAergic neurons in the VTA (Supplementary Fig. 6d,e). (page 10, lines 185-187 in the revised manuscript)

Beyond the inputs to DAergic neurons, understanding how their downstream outputs drive opposing behaviors remains a critical question in the field^{54,55}. A recent study has highlighted the role of distinct DAergic downstream projections in mediating the influence of social interaction on addiction susceptibility: optogenetic activation of the DAergic pathway from the VTA to prefrontal cortex suppressed drug seeking, whereas activation of the VTA-to-nucleus accumbens (NAc) pathway exerted the opposite effect⁵. Moreover, distinct VTA DAergic projections into defined NAc subregions also mediate diverse behavioral functions⁵⁵. Building on these findings, investigating how distinct downstream dopamine signals functionally connect to the Social-reward-Ens^{DA} and Drug-seeking-Ens^{DA} identified in our study will be a key focus of future research. (page 24, lines 489-499 in the revised manuscript)

Supplementary Fig.6 | Spatial distributions of VTA DAergic ensembles activated by social reward or drug seeking. *d*, Representative images showing activated DAergic neuronal ensembles in the VTA during social interaction, with quantification of the proportion of these ensembles relative to the total dopamine neuron population in the VTA. *e*, Representative images showing activated DAergic neuronal ensembles in the VTA during drug relapse, with quantification of the proportion of these ensembles relative to the total dopamine neuron population in the VTA.

4. For Fig 3e, the authors hypothesize that Social-reward-Ens-DA neurons inhibit Drug-seeking-Ens-DA neurons via non-canonical release of GABA and demonstrated a monosynaptic inhibitory connection between these two groups of dopamine neurons. However, this evidence alone is insufficient. Additional experiments such as IF or in situ should be performed to confirm co-expression of GABA markers (e.g., GAD or vGAT) and TH within Social-reward-Ens-DA neurons. Furthermore, the authors should ablate/inhibit all VTA GABAergic neurons to test whether the IPSCs still persist. Lastly, could the authors explain the observed latency in the light-evoked IPSCs?

Response: We thank the Reviewer for this valuable comment. We acknowledge that the initial evidence supporting non-canonical GABA release from Social-reward-Ens^{DA} neurons was insufficient. In response to your suggestions, we have supplemented a series of complementary experiments to address this critical mechanistic question:

We performed immunohistochemical co-labeling experiments to examine the colocalization of TRAPed neurons with Glutamate decarboxylase (GAD, a classic marker for GABAergic neurons). The results revealed that only ~10% of Social-reward-Ens^{DA} colocalized with GAD (**Supplementary Fig. 11a,b**). We further examined the effects of GAD antagonists on IPSCs induced by optogenetic activation of Social-reward-Ens^{DA}. The results revealed that the GAD inhibitor 3-mercaptopropionic acid (3-MPA) had no effect on IPSCs (**Supplementary Fig. 11h**).

These lines of evidence indicate that the protective effect of social reward against drug seeking is not mediated by non-specific labeling of GABAergic neurons by the TH promoter.

To clarify the mechanism underlying the protective effect of social reward against drug seeking, we reviewed relevant literature and found that Aldehyde Dehydrogenase 1a1 (ALDH1a1, a marker that colocalizes with TH and mediates a GABA synthesis pathway in midbrain DAergic neurons) but not GAD mediates a GABA synthesis pathway in midbrain DAergic neurons (Kim et al., 2015, Science; Tritsch et al., 2012, Nature). We therefore sought to determine whether ALDH1a1 drives functional GABA co-release by Social-reward-Ens^{DA}. To this end, we supplemented the following experiments:

We performed immunohistochemical co-labeling experiments to examine the colocalization of Social-reward-Ens^{DA} with ALDH1a1. The results revealed that ~37% of Social-reward-Ens^{DA} colocalized with ALDH1a1 (**Supplementary Fig. 11c,d**). We also examined the effects of the ALDH1a1 inhibitor on IPSCs induced by optogenetic activation of Social-reward-Ens^{DA}. The results revealed that the ALDH1a1 inhibitor 4-(diethylamino)-benzaldehyde (DEAB) significantly attenuated the amplitude of light-evoked IPSCs (**Supplementary Fig. 11i**).

These lines of evidence indicate that GABA co-release from TH-positive neurons, more precisely from those expressing ALDH1a1, mediates the protective effect of social reward against drug relapse.

Finally, regarding the observed latency in light-evoked IPSCs, we attribute this to minor systematic errors in the light stimulation delivery system used in the electrophysiology experiments.

The following text and image have been added to the manuscript:

Given that previous studies demonstrated that Aldehyde Dehydrogenase 1a1 (ALDH1a1) but not the glutamate decarboxylase (GAD) mediates a GABA synthesis pathway in midbrain DAergic neurons^{22,23}, we sought to determine whether ALDH1a1 drives functional GABA co-release by Social-reward-Ens^{DA}. To address this, we examined GAD and ALDH1a1 expression in Social-reward-Ens^{DA} via immunohistochemical co-labeling with TH. Our results revealed that only ~10% of Social-reward-Ens^{DA} expressed GAD, whereas ~37% of these neurons expressed ALDH1a1 (Supplementary Fig. 11a-d). Moreover, treatment with the ALDH inhibitor 4-(diethylamino)-benzaldehyde (DEAB, 10 μ M) but not the GAD inhibitor 3-mercaptopropionic acid (3-MPA, 500 μ M) dramatically reduced light induced IPSC amplitude (Supplementary Fig. 11e-i). Collectively, these findings establish that ALDH1a1 is required for functional GABA co-release by Social-reward-Ens^{DA}.

(page 13-14, lines 265-276 in the revised manuscript)

Supplementary Fig. 11 | Aldehyde Dehydrogenase 1a1 Mediates the inhibitory effect of Social-reward-Ens^{DA} on Drug-seeking-Ens^{DA}. **a**, Representative histology images of Social-reward-Ens^{DA} co-label TH or GAD. **b**, The proportion of co-labeled neurons (TH + Social-reward-Ens^{DA} vs GAD + Social-reward-Ens^{DA}) represents the total Social-reward-Ens^{DA}. **c**, Representative histology images of Social-reward-Ens^{DA} co-label TH or ALDH1a1. **d**, The proportion of co-labeled neurons (TH + Social-reward-Ens^{DA} vs ALDH1a1 + Social-reward-Ens^{DA}) represents the total Social-reward-Ens^{DA}. **e, f**, Strategy for Virus injection. **g**, Schematic of the experimental procedure for electrophysiology recording. **h**, Left: Representative traces of evoked IPSCs upon bath application of GAD inhibitor 3-mercaptopropionic acid (3-MPA, 500 μ M). Right: Amplitudes under control conditions (ACSF) or 3-MPA, normalized to baseline. **i**, Left: Representative traces of evoked IPSCs upon bath application of ALDH inhibitors 4-(diethylamino)-benzaldehyde (DEAB, 10 μ M). Right: Amplitudes under control conditions (ACSF) or DEAB, normalized to baseline (DEAB: $n = 8$ neurons, paired t test, $**p < 0.01$).

5. I'm curious about your experiment being done under social isolation conditions. Would social interaction still inhibit relapse in group-housed mice?

Response: We sincerely appreciate the Reviewer's insightful question regarding the generalizability of our findings to group-housed animals, given that our experiments were conducted under social isolation conditions. We address this concern with three key points:

- 1) Firstly, the social isolation paradigm was necessitated by the technical requirements of our intravenous self-administration surgery. The indwelling dorsal venous catheter used for drug delivery must remain unobstructed throughout the training period, and group housing would pose a high risk of catheter damage due to conspecific interaction. Thus, isolation was a practical necessity to ensure the integrity of our experimental procedures.
- 2) Secondly, multiple published studies have demonstrated that group housing itself functions as a social reward and exerts a protective effect against drug addiction. For example, Alexander et al. (1978), Raz et al. (2010), and Bozarth et al. (1989) reported that group-housed rodents exhibit significantly reduced acquisition of drug self-administration compared to socially isolated controls. These findings align with the core conclusion of our study that social context modulates drug-seeking behavior.
- 3) Finally, to directly address this question, we conducted supplementary experiments in which rats were group-housed after cocaine SA training. Our results showed that sustained social interaction with drug-naïve conspecifics accelerated the extinction of drug-associated memories and effectively suppressed drug relapse (**Supplementary Fig. 18c,d**), further confirming the protective role of social context in addiction.

The following text and image have been added to the manuscript:

Furthermore, social housing environments (with drug-naïve partners), characterized by sustained conspecific interaction, exert a long-term protective influence on drug-seeking behavior. Specifically, such housing conditions accelerated the extinction of drug-associated memory and suppressed drug relapse (Supplementary Fig. 18c,d). (page 21, lines 427-431 in the revised manuscript)

Supplementary Fig.18 | The protective effect of acute social interaction against drug seeking is transient. *c.* Experimental design to investigate the effect of housing conditions (pair-housed vs. socially isolated) on cocaine-seeking behavior. *d.* Number of nose-pokes during cocaine self-administration training, extinction and relapse tests. Pair-housed rats exhibited accelerated extinction ($n = 9-10$, three-way ANOVA with simple effect analysis: $F_{(13,5)} = 4.799$, $p = 0.047$, more statistics see Supplementary Table14B) and reduced cocaine seeking during the relapse test ($n=9-10$, two-way ANOVA with simple effect analysis: $F_{(1,17)} = 14.861$, $p = 0.001$, more statistics see Supplementary Table14C) compared with socially isolated rats. All the data are presented as mean \pm SEM.

6. For Figure 6, considering that AAV1 is both anterograde and retrograde, previous study (Lin et al., 2021) indicates VTADA also projects to DRN, albeit at a low proportion. Could this viral strategy inadvertently label DRN-projecting VTA DA neurons? Please clarify.

Response: We sincerely appreciate the Reviewer's critical comment regarding the potential off-target labeling of DRN-projecting VTA DAergic neurons due to the dual anterograde/retrograde transport properties of AAV2/1. We acknowledge that AAV2/1 may exhibit minimal retrograde tracing activity, which raises the concern that our viral strategy could inadvertently label DRN-projecting VTA DAergic neurons. To rigorously rule out this possibility and clarify the functional specificity of the projection pathway, we supplemented a control experiment to dissect the respective roles of DRN-VTA versus VTA-DRN projections. Specifically, we artificially activated these two pathways separately and examined their effects on drug-seeking behavior. The results, now integrated into the manuscript, showed that activation of DRN-VTA serotonergic projections mimicked the therapeutic effect of social reward in suppressing drug seeking, whereas activation of VTA-DRN dopaminergic projections had no significant impact (**Supplementary Fig. 17a-j**). This directly confirms that our observed protective effect is mediated by DRN-VTA projections (the anterograde pathway we intended to target).

The following text and image have been added to the manuscript:

To rule out the effect of potential retrograde transport of AAV2/1-Cre²⁷ and make our conclusion more convincing, we established control groups to dissect the functional roles of DRN-VTA projections versus VTA-DRN projections. The results showed that activation of the DRN-VTA serotonergic projections, rather than VTA-DRN dopaminergic projections, mimicked the therapeutic effect of social reward against drug seeking (Supplementary Fig. 17a-j), confirming the critical role of DRN serotonergic inputs. (page 19, lines 383-390 in the revised manuscript)

Supplementary Fig. 17 | Activation of the serotonergic DRN-VTA but not the dopaminergic VTA-DRN projections mimicked the therapeutic effect of social reward against cocaine seeking. a, Left: Strategy for Virus injection. Right: Representative photo of hM3Dq-mCherry expression in the VTA. Scale bar: 500 μ m. b,

Experimental scheme for activation of the DRN-VTA serotonergic projections to mimic the therapeutic effect of social reward against cocaine seeking. The rats were intraperitoneally injected with CNO 30 minutes before the cocaine relapse test. **c**, Number of nose-pokes during the 10-day cocaine self-administration training. **d**, Number of nose-pokes during the 14-day extinction training. **e**, Number of nose-pokes during the relapse test, an activation of the DRN-VTA serotonergic projections significantly inhibits cue-induced active nose-pokes, mimicking the effects of social reward on cocaine seeking ($n = 8$, two-way ANOVA with simple effect analysis: $F_{(1,14)} = 5.803$, $p = 0.030$; more statistics see Supplementary Table 8G-I). **f**, Left: Strategy for Virus injection. Right: Representative photo of hM3Dq-mCherry expression in the DRN. Scale bar: 500 μm . **g**, Experimental scheme to investigate the effect of the VTA-DRN dopaminergic projection on cocaine-seeking behavior. **h**, Number of nose-pokes during the 10-day cocaine self-administration training. **i**, Number of nose-pokes during the 14-day extinction training. **j**, Number of nose pokes during the relapse test. Activation of the VTA-DRN dopaminergic projection had no effect on cocaine-seeking behavior during the relapse test ($n = 8$, two-way ANOVA with simple effect analysis: $F_{(1,14)} = 1.163$, $p = 0.299$, more statistics see Supplementary Table 8J-L). All data are presented as mean \pm SEM.

7. What major types of neurons project from DRN to the Social-reward-Ens-DA neurons? Are they DA or 5-HT neurons?

Response: We sincerely appreciate the Reviewer's valuable question regarding the neuronal type of DRN projections to Social-reward-Ens^{DA} (i.e., whether they are DAergic or serotonergic). Based on prior literature and our own experimental findings, we speculated that these projections are primarily serotonergic, supported by the following two key lines of evidence:

- 1) Prior anatomical tracing studies: Lin et al. (2020) demonstrated that DAergic neurons in the DRN mainly innervate the central amygdala (CeA) and bed nucleus of stria terminalis (BNST). In contrast, other studies (Wang et al. (2023), Paquelet et al. (2022)) have confirmed that DRN neurons projecting to the VTA are predominantly glutamatergic or serotonergic, rather than DAergic.
- 2) Functional relevance of serotonergic DRN-VTA projections: Our experimental results show that activation of serotonergic DRN-VTA projections mimics the therapeutic effect of social reward in suppressing drug seeking (**Supplementary Fig. 17a-e**). This aligns with prior works (Li et al., 2021, Miyazaki K et al., 2021) showing that serotonin's role in mitigating compulsive drug-seeking behaviors.

The following text has been added to the manuscript:

The DRN has been classically recognized as a nucleus that contains an anatomically and functionally diverse population of serotonergic neurons⁴⁷, many of which innervate brain areas involved in reward processing⁴⁸. Social interaction, a behavior with high reward salience, has been linked to serotonergic activity in the DRN⁴⁹—a link our findings corroborate and extend. Brain-wide monosynaptic mapping revealed that Social-reward-Ens^{DA} receives substantial inputs from DRN, and social interaction robustly activates DRN populations that project to the VTA⁵⁰. Prior work demonstrates that serotonergic DRN-VTA projections enhance social interaction time in stress-susceptible mice, while silencing this pathway diminishes social interaction time⁵¹. Consistent with this, we found that activation of VTA DAergic neurons that receive input from the DRN or the serotonergic DRN-VTA projections mimicked the therapeutic effect of social reward against drug seeking (Supplementary Fig. 17a-e). These findings provide partial mechanistic support for models positing serotonin's role in mitigating compulsive drug-seeking^{52,53}. (page 23, lines 475-488 in the revised manuscript)

8. How do these two populations of neurons respond to other types of rewards, such as tasty food or addictive drug injections?

Response: We sincerely appreciate the Reviewer's insightful question regarding the responses of Drug-seeking-EnsDA and Social-reward-EnsDA to other reward types (e.g., tasty food, addictive drugs). To address this, we conducted supplementary experiments by artificially activating these two ensembles and examining their effects on feeding behavior. Our results showed that activation of Drug-seeking-EnsDA significantly inhibited feeding behavior within 2 hours, while activation of Social-reward-EnsDA had no significant impact on feeding (Supplementary Fig. 19e-h). This finding highlights the functional specificity of the two ensembles in processing different rewards.

The following text and test have been added to the manuscript:

Moreover, we found that activation of Drug-seeking-Ens^{DA} significantly inhibited feeding behavior (Supplementary Fig. 19e-h), further supporting the competitive functional relationship between natural rewards and drug-seeking behavior. (page 24, lines 507-510 in the revised manuscript)

Supplementary Fig.19 | Comparison of therapeutic effects between social and sucrose reward on cocaine seeking. e, Schematic of the experimental design for refeeding following activation of Social-reward-Ens^{DA}. f, Comparison of cumulative food intake in fasted rats. Rats received i.p. injections of CNO or vehicle 30 min prior to being given free access to food. g, Schematic of the experimental design for refeeding after activation of Drug-seeking-Ens^{DA}. h, Comparison of cumulative food intake in fasted rats. Rats received i.p. injections of CNO or vehicle 30 min prior to being given free access to food. Fasted rats administered CNO consumed significantly less food than those administered vehicle during the first 1 and 2 h of refeeding ($n=8$, two-way ANOVA with simple effect analysis: $F_{(2,13)}=5.545$, $p = 0.018$, more statistics see Supplementary Table15B). All the data are presented as mean \pm SEM.

Minor Issues:

1) For Fig 2e, 2f, the statistical analysis should be presented as percentages rather than as raw numbers.

Response: We sincerely appreciate the Reviewer's constructive suggestion. We have revised the statistical analysis in Figures 2e and 2f, replacing raw numbers with percentages to enhance clarity and interpretability of the data.

The following text and image have been added to the manuscript:

Notably, the proportion of co-labeled DAergic neurons across social reward and drug seeking conditions was significantly lower than that observed in control groups subjected to the same stimulus twice (Fig.2e,f), suggesting largely distinct neuronal populations for each condition. (page 9, lines 169-172 in the revised manuscript)

Fig.2 | Distinct VTA DAergic ensembles respond to social reward and drug seeking. e, Proportion of neurons co-labeled for Label 1 and Label 2 (n = 4-5; one-way ANOVA with Bonferroni multiple comparisons test, **p < 0.01). f, Proportion of neurons co-labeled for Label 1, Label 2, and TH (n = 4-5; one-way ANOVA with Bonferroni multiple comparisons test, **p < 0.01).

2) For the miniscope data, the n-value should not only represent the number of neurons but also indicate how many animals these neurons are from.

Response: We sincerely appreciate the Reviewer's constructive suggestion. We have revised the presentation of n-values for all miniscope imaging data in the manuscript.

3) In Fig 5c, the starting cell is unclear, and the background appears messy.

Response: We sincerely appreciate the Reviewer's careful observation. We have revised Figure 5c to clarify the starting cell and clean up the messy background, improving the clarity and readability of the image.

The following image has been revised to the manuscript:

Fig.5 | Social-reward-EnsDA and Drug-seeking-EnsDA in the VTA establish different connection patterns. c, Representative images of the VTA injection site and magnified view of starter cells with tamoxifen injection. Scale bar: 200 μ m.

Response letter to manuscript NCOMMS-25-79000A

The authors express their gratitude to the Reviewers for providing valuable feedback. Detailed responses addressing each comment are provided below. Comments from the Reviewers are highlighted in **bold**, while author responses are presented in **plain blue text**. Modifications made to the manuscript are indicated by *italicized text on a yellow background*. Page and line numbers referenced in the responses correspond to those found in the revised manuscript.

Reviewer #1 (Remarks to the Author)

The authors provided a very thorough set of good responses to my concerns. I am fine with the revised version now with one exception:

In response to my point 9, the authors quote a review by Yuste et al 2024 for their definition of a neuronal ensemble -“a population of neurons involved in a particular computation”. This is only a partial definition that does not distinguish between neuronal ensemble versus cell type, and is thus insufficient. When talking about neuronal ensembles as an encoding unit, Yuste is referring to a set of neurons that are selected by a learned set of cues, and others have found are composed of multiple cell types acting together. Neuronal ensembles are not mere cell types that are involved differentially in different rewards (e.g. social versus drug reward) or (e.g. nicotine versus cocaine reward). For example in cortex, different ensembles encoding different information often have the same relative percentages of cell types.

What I was trying to say in my initial review is that you are most likely looking at different cell types for social versus drug reward that are genetically predetermined prior to learning - this means they are not neuronal ensembles in the more meaningful and informative sense. Ensembles are about encoding high-resolution learned information, and not about encoding low-resolution largely innate information that can be handled by mere cell types.

Equating neuronal ensembles with cell types is due to an incorrect over-generalization of the term neuronal ensembles. Unfortunately, this is an overly common error in the field. Please fix and I am fine with the rest of the paper, which was very interesting and a lot of good work.

Response: We sincerely appreciate the reviewer’s insightful and constructive comments. We apologize for our earlier misunderstanding of the reviewer’s point, and we now understand the concern that we are most likely looking at different cell types for social versus drug reward that are genetically predetermined prior to learning. We acknowledge the reviewer’s critical distinction that neuronal ensembles encode high-resolution learned information, rather than low-resolution largely innate information that can be accounted for by cell types alone.

1. Regarding the drug-relapse-ensemble, our paradigm involved extensive operant conditioning of nosepoke responses, cue presentations, and drug reward. We defined neurons responsive to nosepoke as the drug-relapse ensemble, which explicitly indicates that this ensemble is not pre-existing prior to learning. For the social-reward-ensemble, based on your suggestion, we have added experiments involving operant conditioning for social nosepoke, social cues, and social reward in the previous round of revision. These additional data further demonstrate

that the social-reward ensemble is also established through learning.

2. Although our results support that both ensembles are experience-dependent, we agree that we cannot completely rule out the contribution of genetically predetermined cell-type differences underlying social versus drug reward. Accordingly, we have revised the relevant interpretations. We thank the reviewer again for this important correction, which has greatly improved the clarity and accuracy of our manuscript.

The following texts have been revised to the manuscript:

Neuronal ensembles are defined as populations of neurons that encode specific information through coordinated activity patterns induced and shaped by learning²¹. (page 9, lines 173-174 in the revised manuscript)

To address this alternative explanation, we trained rats to perform operant responses for both social self-administration and cocaine self-administration, then assessed whether distinct neuronal ensembles shaped by learning were engaged when rats executed goal-directed behaviors to obtain these two discrete reward-related stimuli (Supplementary Fig. 9a,b). Our results demonstrated that only 10% of recorded neurons (26 out of 261 neurons; n = 2 animals) exhibited dual responsiveness to both social and cocaine cues (Supplementary Fig. 9c-p). Subsequent statistical analyses confirmed a lack of significant overlap between social-seeking- and drug-seeking-responsive neuronal populations (Supplementary Fig. 9q). Collectively, these findings establish that VTA DAergic populations exhibit stable, stimulus-specific identities and that their segregation is not a consequence of temporal factors or behavioral confounds, consistent with the properties of learning-induced neuronal ensembles. (page 11-12, lines 224-235 in the revised manuscript)

We speculate that this reciprocal interaction is progressively established during repeated drug use, although we cannot fully exclude the existence of genetically predetermined cell types in the VTA; further investigation will be needed to dissect these possibilities. (page 23, lines 470-473 in the revised manuscript)

Reviewer #2 (Remarks to the Author):

I have no further comments

Response: Thank you for your review and positive feedback on our manuscript. We highly appreciate your time and efforts in evaluating our work.

Reviewer #3 (Remarks to the Author):

The authors answered all my questions and I have no further comments.

Response: Thank you for your review and positive feedback on our manuscript. We are glad that we have addressed all your questions satisfactorily, and we greatly appreciate your time and effort in evaluating our manuscript.